# Investigating sustainable development in transportation enterprises: Novel insights from new institutional economics and human capital theory. Evidence from HCM, Vietnam

Vu Thi Kim Hanh[1,2*☉], Nguyen Hong Nga[1,2*☉]

**1** Faculty of Economics, University of Economics and Law, Ho Chi Minh City, Vietnam, **2** Faculty of Economics, Vietnam National University, Ho Chi Minh City, Vietnam

☉ Both authors contributed equally to the conception and design of the study.
* hanhvtk20702@sdh.uel.edu.vn (VTKH); nganh@uel.edu.vn (NHN)

## Abstract

This study examines the statistical associations between institutional factors, human capital, and sustainable transportation development in urban transport hubs within developing economies. Guided by North's institutional theory and human capital theory, it investigates how institutions and human capital are statistically associated with the social, economic, and environmental dimensions of sustainable transportation development, emphasizing statistically significant direct effects. Data were obtained from 354 transportation enterprises in Ho Chi Minh City, Vietnam, via stratified probability sampling and analysed using partial least squares structural equation modelling. Results indicate that institutional factors exhibit both positive and negative statistical associations across the three dimensions, while the social and environmental dimensions are reciprocally associated with institutional factors in positive and negative ways, respectively. Likewise, human capital exhibits positive, statistically significant bidirectional relationships with the social and environmental dimensions, highlighting mutual reinforcement. By documenting these two-way relationships, the study advances theory and provides applied insights. In particular, it highlights the value of aligning regulations with enterprise needs and investing in human capital to guide policies that promote institutional effectiveness and sustainable urban mobility.

## 1. Introduction

Urban transport hubs in both developing and developed regions are increasingly confronted with sustainability pressures stemming from accelerated urban growth and expanding economic activity. The transport sector, in particular, experiences heightened demands due to rising energy use, environmental strain, and greenhouse gas emissions [1–3]. Serving as a central driver of economic connectivity, transportation

**Data availability statement:** All data underlying the findings described in this manuscript have been uploaded as supplementary information.

**Funding:** This research is funded by University of Economics and Law, Vietnam National University Ho Chi Minh City, Vietnam. No grant number is available, and no individual author received a grant or salary support from this funding. The specific roles of this author are articulated in the 'author contributions' section. The funders had no role in study design, data collection and analysis, decision to publish, or preparation of the manuscript.

**Competing interests:** The authors have declared that no competing interests exist.

plays a pivotal role worldwide, especially amid global integration and the pursuit of SDGs [4]. Recent studies emphasize the importance of energy-efficient approaches and cross-disciplinary strategies in transportation planning to achieve both environmental and economic objectives within smart urban frameworks [5]. Building on this perspective, the present study investigates how institutional arrangements and human capital contribute to sustainable transportation outcomes in rapidly urbanizing areas.

Despite its importance, the transportation sector significantly contributes to negative externalities, such as air pollution, climate change, and resource depletion. Consequently, transport enterprises must balance economic gains with environmental stewardship and social responsibility [6,7]. Sustainable transportation development is vital in mitigating carbon emissions and improving energy efficiency [1,8]. Institutions are key in guiding economic activities by establishing rules and incentives that impact resource allocation, innovation, and firm profitability. They further affect social and environmental outcomes through labour regulations, community expectations, and environmental policies. For instance, institutional frameworks facilitate the adoption of technologies such as real-time shipment tracking, which can improve both operational efficiency and sustainability [9].

Equally critical is human capital, which drives innovation and adaptability. Skilled and environmentally conscious employees support green standards, integrate advanced technologies, and foster continuous improvement within transportation enterprises [10–13]. Institutions and human capital together provide a foundation for sustainable transportation, particularly in the face of global environmental challenges and local constraints.

However, translating these foundations into tangible outcomes remains difficult in emerging urban contexts due to limited resources, regulatory inconsistencies, and rapid development. While many sustainability tools and frameworks exist, real-world implementation requires data-driven evaluations of key indicators [2,3,6,14]. Research also highlights the importance of ecological awareness and environmentally responsible behaviour in fostering sustainability-oriented perspectives [8]. Sustainable transportation is no longer optional but a necessary strategy amid global calls for emission reductions, renewable energy use, and energy efficiency improvements, particularly in urban freight and public transport systems [15–16]. Global frameworks such as the Paris Agreement [17] and the SDGs [4] have accelerated this shift by mandating decarbonization and infrastructure resilience [18–20]. These serve as both external pressures and strategic roadmaps for transport enterprises seeking long-term sustainability.

The COVID-19 pandemic added urgency to this transition by exposing vulnerabilities in mobility systems and disrupting urban logistics [21–22]. However, it also presented opportunities to reimagine transport systems as more sustainable and resilient [23]. Pandemic-driven changes in travel behaviour, remote work, and public awareness of health-environment linkages offer momentum for transportation enterprises to adopt sustainability strategies [24]. These approaches yield benefits such as reduced emissions and costs, and enhanced urban resilience and public health [25].

Embedding sustainability at the transportation enterprise level is thus strategically vital, especially in rapidly urbanizing areas.

Vietnam represents a notable development success. Since the 1986 "Đổi Mới" reforms, it has transformed from one of the poorest nations into a vibrant lower-middle-income economy. GDP per capita rose from below $700 in 1986 to nearly $4,500 in 2023. Human capital has also improved, with lower-secondary enrolment at 95% and upper-secondary at 80% in 2024. With a learning-adjusted schooling average of 10.2 years, Vietnam ranks second in ASEAN after Singapore and leads its lower-middle-income peers. However, institutional and sustainability issues remain, particularly in transportation [26]. Ho Chi Minh City (HCMC) is selected as the research site for its economic significance and pressing transport concerns, being the nation's largest financial and commercial hub, it contributed around 16% to national GDP in 2023. Its 2024 gross regional domestic product rose by 7.17%, leading nationwide growth [27]. From 2010 to 2023, the city consistently contributed 16–24% of national GDP [27–28]. However, it faces severe congestion, pollution, and emission issues. Motorcycles are the primary source of CO and account for about 50% of $CO_2$ emissions [29]. Air quality is heavily influenced by $NO_2$ and temperature in the morning, and elevated CO and $O_3$ concentrations in the afternoons, primarily resulting from heavy traffic [30]. In January 2025, pollution levels in the city were recorded at over 11 times the WHO threshold, ranking it among the most polluted in Southeast Asia [31].

This study investigates the bidirectional statistical associations across institutional elements, human capital, and the social, economic, and environmental pillars of sustainable transport development. Drawing on institutional theory, it differentiates between enabling and constraining institutional aspects, such as regulations, policies, and societal norms, and examines their statistical associations with sustainability outcomes. Concurrently, drawing on human capital theory, it examines how human capital, such as knowledge, skills, and competencies, are statistically associated with these sustainability dimensions. Importantly, the study also assesses how each sustainability dimension is statistically associated with institutional factors and human capital. HCMC serves as an illustrative context for analysing these interrelationships within a resource-constrained, rapidly urbanizing setting.

The structure of the paper is as follows. Section 2 reviews prior studies on sustainable transportation alongside the theoretical foundations of institutional and human capital approaches, highlighting research gaps, contributions, hypotheses, and the conceptual model. Section 3 describes the research design, including context, sampling, data collection, measurement instruments, and the rationale for applying PLS-SEM. Section 4 reports the empirical results, followed by Section 5, which discusses key implications and contributions. Section 6 offers concluding remarks and practical guidance, and Section 7 addresses study limitations and avenues for future inquiry.

## 2. Literature review

### 2.1. Advancing sustainable transport: Key dimensions, innovative approaches, and strategic significance

Sustainable transportation is now a central focus within both global and national policy frameworks, aiming to balance environmental protection, economic growth, and social equity [2,3,6,14]. In rapidly urbanizing developing economies, transportation systems must address increasing complexity and sustainability expectations [1]. Beyond reducing emissions, sustainable urban mobility requires integrated strategies across economic, social, and environmental dimensions [3,4,7,32,33]. This study adopts a three-dimensional framework (Figure 1), emphasizing profitability and productivity (economic), equitable access (social), and ecological mitigation through greener options like urban transit systems and autonomous transport (environmental).

Performance metrics include carbon emissions, energy use, and inclusive access [40]. In cities like HCMC, challenges such as congestion, informal transit, and fragmented infrastructure demand institutional reforms, digital governance, and human capital development aligned with the United Nations' SDGs [2,3,6,7,14,29–31].

Mobility as a Service, which digitally integrates various modes (e.g., buses, ride-hailing, bike-sharing), offers a solution to reduce car dependency and improve access [34], though its implementation is hindered by infrastructural and policy

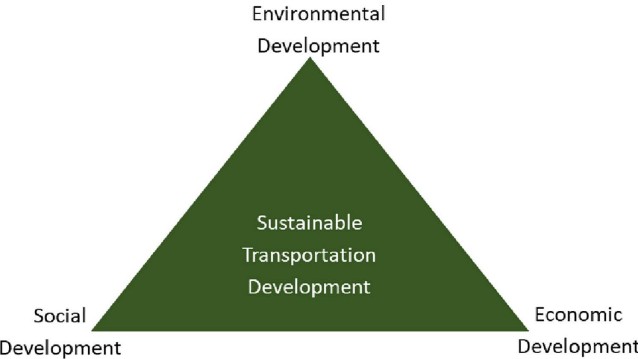

**Fig 1. Framework for Sustainable Transportation Development.** Source: modified from [2,3,6].

barriers [35–36]. Shared transport options like Grab and Gojek contribute to congestion reduction, though their sustainability varies by trip type and local energy mix [37–38]. Electric vehicles and autonomous technologies offer reductions in fossil fuel use and safety improvements [39]. Smart transport systems leveraging artificial intelligence, big data analytics, and the internet of things further optimize routing and logistics, particularly in congested cities, provided that institutional and regulatory support is present [1,40,41]. Intermodal transport, integrating road, rail, and waterways, is also vital. Studies show it can reduce emissions by achieving 77.4% higher efficiency and enhance energy savings by 43.48% compared to single-mode approaches [42]. For firms facing institutional and human capital limitations, intermodal solutions provide economical routes toward sustainability [43–45].

This research investigates the bidirectional relationships between institutional structures, human capital, and the economic, environmental, and social dimensions of sustainable transportation development within the context of an urban transport hub in a developing economy. It explores the ways in which institutional conditions and human capital attributes are associated with the uptake and execution of sustainable transport practices, while also analysing how evolving priorities across the three sustainability dimensions are associated with changes in institutional arrangements and contribute to human capital advancement.

### 2.2. Integrating institutional and human capital theories: A dual-lens approach to sustainable transportation development

According to North's institutional theory, institutions represent the formal and informal "rules of the game" that shape economic and social interactions [46]. They encompass elements such as legal frameworks, regulatory systems, governance arrangements, and prevailing social norms [9]. Rather than reiterating definitional aspects, this study adopts a functional and dynamic view of institutions as evolving structures that both influence and are reshaped by enterprise behaviour over time. This bidirectional interaction is particularly salient for transportation enterprises in emerging economies, where institutional frameworks condition strategic choices, resource mobilization, and sustainability engagement. New Institutional Economics adds that institutional change aligns with economic incentives and governance reform. For instance, Nordic road management liberalization followed deliberate restructuring [47], and institutions shape entrepreneurship via networks and norms influencing opportunity recognition [48]. Institutions exert interconnected influences on the economic, social, and environmental dimensions of sustainable transportation. Economically, stable and coherent institutional frameworks reduce transaction costs, foster investment, and promote efficient market behaviour [9,46,49]. Socially, institutions shape labour laws, equity norms, and public engagement, all of which affect fairness and inclusion in transport systems [9,50–52]. Environmentally, legal and regulatory structures provide mechanisms to enforce emission standards, support

sustainable infrastructure, and manage ecological impacts [9,50–52]. These effects are observable across multiple contexts: institutional stability has supported rural tourism development in Lithuania [49]; legal frameworks have facilitated the sustainable development of mining enterprises and stakeholder value creation in Europe [53]; and macroeconomic and regulatory systems have enabled the long-term viability of sustainable development in transportation companies in the Eastern European Union [6]. Conversely, institutional weaknesses, such as fragmented regulations, increased tax burdens, bureaucratic inefficiencies, and informal barriers, pose serious constraints. For instance, small firms in Russia face institutional plunder and entry barriers [54], while in Slovenia, regulatory bureaucracy and high capital costs remain key limitations [55]. In such contexts, misaligned institutional incentives may even foster unproductive entrepreneurship [56], and siloed governance often delays sustainability transitions in urban transport [57]. Institutions continue to evolve under competing pressures and varied cultural settings. Studies show how institutional arrangements interact with technological governance in international trade and metropolitan transport in the United States [50]. Yet, while institutions often support economic success, their role in enabling sustainable development remains inconsistent across contexts [51]. Localized analyses from Melbourne (Australia) and Vancouver (Canada) illustrate how institutional path dependencies can either delay or enable adaptive urban transport reforms in response to shifting mobility demands [52,58]. In Nordic countries, targeted institutional reform, such as the liberalization of road management, has contributed to more effective transport governance [47].

While institutions structure the external operating environment, Becker's Human Capital Theory offers a complementary internal perspective, focusing on the knowledge, skills, and attributes embedded in individuals [59]. These attributes, enhanced through education, training, and health investments, are not only drivers of productivity but, in sustainability-focused contexts, are crucial for fostering social inclusion, environmental responsibility, and digital transformation. In emerging urban economies such as HCMC, human capital serves as both a foundation and an amplifier of sustainable transportation. It enables green innovation, enhances workforce adaptability, and increases responsiveness to institutional signals. In turn, sustainable systems contribute to human capital development by promoting equitable employment, upskilling, and access to clean technologies [59]. Despite its strategic importance, a persistent gap remains between the theoretical promise of human capital and its practical integration into enterprise-level sustainability initiatives. This underscores the managerial imperative to establish a well-structured and efficient flow of information within transportation enterprises [60]. Empirical evidence from Austria's road freight sector further highlights the pivotal roles of employee engagement and organizational culture in advancing sustainability outcomes [61]. Similarly, in Vietnam's transport sector, digital literacy and employee motivation have been identified as essential enablers for converting external pressures, such as the rise of the sharing economy, into tangible progress toward sustainable development goals [62]. Furthermore, data from the Organisation for Economic Co-operation and Development reveal that labour productivity improvements within subsectors, rather than labour reallocation, are the principal drivers of efficiency in the transport sector [63]. These findings align with Becker's assertion that investment in a skilled and adaptable workforce serves as a vital complement to physical capital in achieving long-term efficiency and development objectives [59]. Human capital also plays a key role in operational strategies such as agility, cost management, and quality assurance, particularly in firms employing outsourcing models. However, these effects vary by firm size and capacity, presenting challenges for Small and Medium-sized Enterprises (SMEs) [64]. Knowledge and skill management are closely linked to competitiveness and operational efficiency. In tightly regulated, resource-constrained contexts such as HCMC, these capabilities are not optional but essential. Beyond the transport sector, Vietnamese start-up research further supports this trend, showing how entrepreneurial human capital, embodied in education, experience, and digital competency, contributes to innovation and resilience under uncertainty. Moreover, government-led training initiatives and efforts to enhance digital transformation readiness further underpin this developmental trajectory [65–66].

While North's institutional theory and Becker's human capital theory offer distinct lenses through which to examine sustainable transportation development, their intersection provides valuable insights into how people and systems co-evolve

   

within an urban transportation context. Institutions shape the formal and informal rules that govern workforce development, skill utilization, and knowledge transfer [9,46]. At the same time, human capital serves as a key mechanism through which institutions are interpreted, challenged, or reinforced within transportation enterprises [60–62]. For example, in contexts where institutions offer clear regulatory incentives or targeted training programs, human capital development is often accelerated [53,59,65]. Conversely, institutional voids, such as regulatory inefficiencies, excessive tax burdens, or bureaucratic inertia, can hinder the effectiveness of even highly skilled personnel, acting as persistent barriers to innovation and enterprise growth [54–55]. In return, a well-educated and motivated workforce can promote institutional evolution by advocating for transparency, innovation, and reform [62].

These interactions are particularly salient in emerging economies where post-pandemic recovery, digital transformation, and sustainability pressures collide. Human capital and institutional quality do not operate in isolation; rather, their dynamic interplay drives or hinders progress across the economic, social, and environmental dimensions of sustainability. For instance, skilled workers are better equipped to interpret and adapt to changing regulations, while institutions can either enable or restrict the capacity of organizations to attract, train, and retain such talent [2 4,63,67,68].

This integrated perspective reinforces the bidirectional hypothesis framework of the present study, suggesting that sustainable transportation development is both a product and a driver of human and institutional evolution. By explicitly examining these interactions, the study contributes a more holistic understanding of the co-dependent roles of institutional frameworks and human capital in shaping sustainable transport futures.

**Research gap.** Although prior research highlights institutional enablers such as macroeconomic stability and supportive legal frameworks [6,49,53], it also identifies constraints including regulatory red tape, fragmented governance, and informal institutional weaknesses [54,57]. At a broader scale, institutional dynamics affect transport systems through technology adoption, supply chain integration, and equity [50–51]. However, much of the existing literature tends to portray institutions as either supportive or obstructive, thereby neglecting their dual nature and bidirectional relationship with sustainable transportation development. Moreover, few studies examine this complexity within transportation enterprises in post-pandemic urban centres of emerging economies, where adaptive institutional capacity is increasingly critical.

Similarly, despite increasing attention to the role of human capital in sustainability [60–63], several gaps persist: (1) limited research links human capital with all three sustainability dimensions, social, economic, and environmental; (2) the reciprocal associations between sustainability outcomes and human capital attributes such as leadership, creativity, and skills remain underexplored; (3) integrated investigations within resource-constrained, post-pandemic urban transport sectors in emerging economies are notably lacking.

These parallel research gaps highlight the need for an integrated theoretical approach. Drawing from North's institutional theory, this study conceptualizes institutions as comprising multiple formal and informal dimensions, such as regulatory transparency, enforcement effectiveness, cultural norms, and bureaucratic efficiency. Rather than labelling institutions as strictly 'supporting' or 'challenging,' we examine how each component differentially shapes sustainable practices in transportation enterprises. However, for modelling and empirical purposes, these multifaceted institutional elements are grouped under broadly facilitative or constraining influences. Building on Becker's human capital theory, this study moves beyond the conventional treatment of institutions and human capital as isolated constructs, instead conceptualizing them as interdependent and mutually constitutive in shaping transportation sustainability outcomes. Institutional factors, whether constraining (e.g., regulatory burdens, corruption, restrictive rules, weak norms, limited support, and high compliance costs) or supportive (e.g., enabling regulations, adaptive norms, informal collaboration, and constructive conventions), are associated with the three dimensions of sustainable development in contexts of limited human capital. Conversely, human capital, through leadership, knowledge sharing, engagement, expertise, skill development, satisfaction, and creativity, is positively associated with these same sustainability dimensions in contexts shaped by complex institutional environments. By investigating these reciprocal associations, this study addresses the lack of theoretical integration,

proposing a unified framework to analyse how interactions between institutions and human capital shape transportation sustainability, and how, in turn, sustainability outcomes reinforce institutional and human capital characteristics within transportation enterprises.

**Main contributions.** By integrating institutional and human capital theories within a unified framework, this study provides a holistic perspective on sustainable transportation development in emerging urban economies. It examines how both constraining and enabling institutional factors are associated with economic, social, and environmental development, and how these sustainability dimensions are, in turn, associated with institutional structures. Additionally, the study analyses how human capital characteristics are not only associated with but also reciprocally associated by these three dimensions. Crucially, it addresses the overlooked interplay between institutions, human capital, and sustainability dimensions, offering empirical insights specific to post-pandemic, resource-constrained urban contexts. This contributes to theoretical integration, methodological clarity, and policy relevance in the field of sustainable transportation enterprises.

**Proposed hypotheses.** *H1: Institutional factors (supporting and challenging) are directly associated with the three pillars of sustainable transport*

H1(a1): Challenging institutional factors are negatively associated with the social dimension.
H1(a2): Supporting institutional factors are positively associated with the social dimension.
H1(b1): Challenging institutional factors are negatively associated with the economic dimension.
H1(b2): Supporting institutional factors are positively associated with the economic dimension.
H1(c1): Challenging institutional factors are negatively associated with the environmental dimension.
H1(c2): Supporting institutional factors are positively associated with the environmental dimension.

*H2: The three pillars of sustainable transport are directly associated with institutional factors.*

H2(a1): The social dimension is negatively associated with challenging institutional factors.
H2(a2): The social dimension is positively associated with supporting institutional factors.
H2(b1): The economic dimension is negatively associated with challenging institutional factors.
H2(b2): The economic dimension is positively associated with supporting institutional factors.
H2(c1): The environmental dimension is negatively associated with challenging institutional factors.
H2(c2): The environmental dimension is positively associated with supporting institutional factors.

*H3: Human capital is directly associated with the three pillars of sustainable transport.*

H3(a): Human capital is positively associated with the social dimension.
H3(b): Human capital is positively associated with the economic dimension.
H3(c): Human capital is positively associated with the environmental dimension.

*H4: The three pillars of sustainable transport are directly associated with human capital.*

H4(a): The social dimension is positively associated with human capital.
H4(b): The economic dimension is positively associated with human capital.
H4(c): The environmental dimension is positively associated with human capital.

## 2.3. Framework and study mode

Fig 2 presents the PLS-SEM framework illustrating the reciprocal connections between institutional elements, human capital, and sustainable transportation development, covering social, economic, and environmental dimensions. The model shows how institutional settings and human capital are statistically associated with sustainability outcomes, which in turn are statistically associated with changes in institutional structures and the enhancement of human capital within transportation enterprises.

**Note on Causal Inference:** This study proposes bidirectional hypotheses grounded in theory and prior empirical work; however, the use of cross-sectional data and PLS-SEM modelling does not permit definitive causal conclusions.

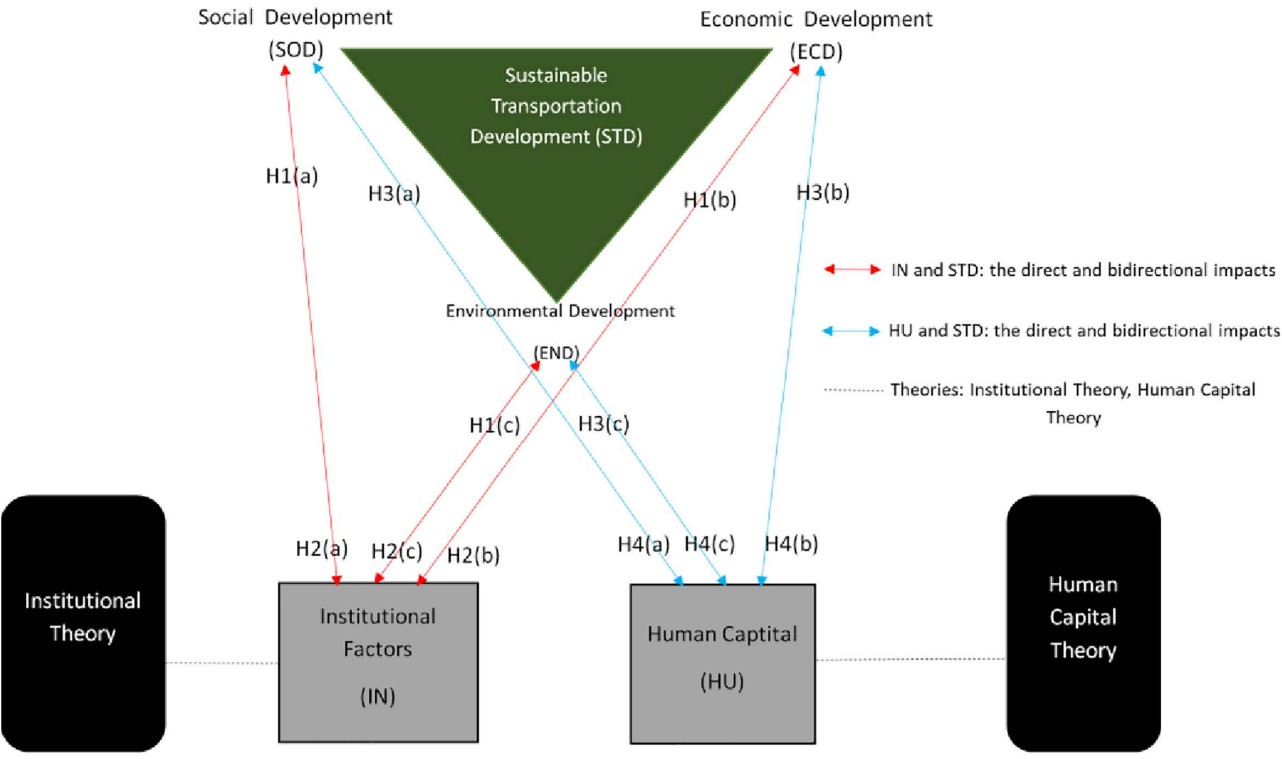

**Fig 2. Framework and study model.** Source: Derived from Institutional Theory [9, 46], Human Capital Theory [59], and related studies [2 - 3, 6].

## 3. Methodology

### 3.1 Research context and sampling strategy

This study examines how institutional dynamics and human capital are associated with environmental, economic, and social sustainability outcomes among transportation enterprises in HCMC. As a rapidly urbanizing metropolitan area facing regulatory complexity, labour shortages, and infrastructure limitations, the city provides a high-stakes environment for investigating real-world sustainability practices. These intersecting challenges make it a compelling setting for assessing both enabling and constraining factors in enterprise-level sustainability efforts.

The research targets operationally active transportation enterprises to ensure that participants are engaged in ongoing transport-related activities. This focus allows for the capture of meaningful interactions among institutional conditions, workforce deployment, and sustainability practices. All transport firms were considered, regardless of profitability, to capture a broad view of sector participation in sustainable development. In 2022, HCMC had 12,025 registered transport firms [27]. Although newer statistics (2023–2025) were unavailable when the study was designed, the 2022 data provided the most reliable baseline. To address possible concerns about outdated figures, the analysis focused on structural and operational features, which tend to remain stable in regulated sectors like transportation. Using stratified probability sampling, a core sample frame of 3,908 active firms was identified. District-level inclusion required at least 2% of the city's active transport enterprises, resulting in a sample frame covering 18 districts. This ensured exclusion of low-activity districts while preserving institutional and geographic diversity.

In each district (stratum), simple random sampling ensured equal selection chances for eligible firms. This two-stage approach, stratification by geography followed by intra-stratum randomization, ensured both representativeness and

methodological rigor. The sample necessarily leans toward enterprises demonstrating continuity and resilience, aligning with the study's aim to explore how functioning businesses engage with sustainability under institutional and human capital constraints. Aligned with PLS-SEM guidelines, the target sample was 300 to ensure model reliability and statistical soundness [69–70]. Participants were senior executives, owners, or managers overseeing both strategic and operational functions. Their perspectives shed light on how transportation enterprises in an emerging urban economy address sustainability challenge.

## 3.2. Data collection and bias control

The survey was carried out in two rounds, August – October 2023 and May – June 2024, covering 18 targeted districts of HCMC. The two-stage data collection approach was adopted intentionally to enhance the robustness of the dataset. The first phase enabled preliminary identification of key trends and potential data inconsistencies, while the second phase allowed for refinement of survey administration and targeted follow-ups. This staggered process also allowed the study to account for potential differences in institutional environments, human capital conditions, and operational realities, which often fluctuate in fast-changing regulatory and business contexts such as those found in HCMC. The strategy not only improved response rates and district-level representativeness but also strengthened the validity of inferences regarding the dynamics of sustainable development in transportation enterprises. A total of 620 survey forms were emailed, following [71], with reminder calls to boost participation. The target was a 48% district-level response rate; actual results reached 60%, yielding 354 valid responses for analysis.

The process went beyond collecting figures, it involved active engagement with transport firms shaping HCMC's mobility sector. This meant building networks, outlining research objectives, and encouraging contributions from owners, executives, and managers who could explain their firms' responses to institutional pressures, human capital challenges, and sustainability aims. To enhance depth, two interview rounds were held with key informants: 60 in the first to identify emerging themes, and 30 in the second to clarify and explore unresolved issues. In total, 90 interviews provided rich insights into how these enterprises navigate regulatory complexity while pursuing sustainable development.

To enhance the credibility of the qualitative results, nonresponse bias was tested using the extrapolation technique, consistent with established recommendations [72–73]. This method assumes late responders resemble nonrespondents in behaviour. Statistical comparisons between early and late groups revealed no significant differences in core variables, suggesting the sample is representative and unaffected by nonresponse bias. To address common method bias, the survey incorporated several controls: clear, neutral wording per [74] to reduce social desirability influence; assurances of confidentiality and anonymity; and collection from knowledgeable informants best positioned to answer the questions, thus avoiding uninformed responses. Post hoc checks, including Harman's single-factor test and marker variable analysis [75], confirmed minimal method variance.

These procedural and statistical measures ensured a reliable dataset for analysing how active transportation enterprises in HCMC navigate institutional pressures and human capital factors in advancing sustainable development.

## 3.3 Sample size justification

To assess whether the sample size of 354 transportation enterprises is sufficient for generalization and valid structural modelling, this study adopts two widely accepted guidelines from the PLS-SEM literature, as recommended by [70].

First, the "10-times rule" suggests that the minimum sample size should be at least ten times the maximum number of arrows pointing at a single latent variable in the PLS path model. In this study, the most complex construct receives three incoming paths, from either human capital, supporting institutions, and challenging institutions toward each sustainable development dimension, namely, social, economic, environmental. Accordingly, the minimum required sample size based on this rule is $10 \times 3 = 30$ cases, which is substantially exceeded by the 354 valid responses obtained. Second, the inverse square root method provides a more conservative and power-sensitive estimate. Assuming a medium path coefficient of

0.2 and a significance level of 5%, the required minimum sample size is calculated as: $n_{(min)} >= (t/r)^2 = (2.486/0/2)^2 = 154.5$. Therefore, a minimum of 155 respondents is required.

To further support the adequacy and representativeness of the sample size in this study, a review of selected empirical research employing PLS-SEM was conducted and is presented in Table 1. These studies span a range of sectors, including SMEs in the transportation sector (transportation SMEs), general SMEs, construction, healthcare, finance, and manufacturing, with sample sizes ranging from as low as 64 to as high as 350 respondents. For example, [76] modeled e-bidding participation using a sample of only 64 construction clients and organizations, while [77] analysed luxury consumption using 157 consumer responses. Similarly, [78] explored the impact of capital structure with 220 business owners in Mexico, and [79] examined employee well-being with 302 organizational respondents in India. Notably, several studies have employed sample sizes that are either smaller than or equivalent to the 354 responses used in the current research. For instance, [80] conducted PLS-SEM analysis using 305 participants. The number of structural paths pointing to a single latent construct in these models ranges between 3 and 9, demonstrating that sample sizes within the 150–350 range are methodologically acceptable and commonly utilized for reliable model estimation and hypothesis testing in PLS-SEM.

**Table 1. Comparative Overview of Sample Sizes in Selected Empirical Studies Using PLS-SEM Across Various Fields.**

| No | Study title | No. of Paths Targeting a Single Latent Variable | Sample size | Respondent Type and Country | Reference |
|---|---|---|---|---|---|
| 1 | E-commerce adoption by SMEs and its effect on marketing performance: An extension of the TOE framework with AI integration, innovation culture, and customer tech-savviness | 5 | 305 | SMEs, Palestine | [80] |
| 2 | Determining factors related to artificial intelligence (AI) adoption among Malaysia's SMEs businesses | 5 | 196 | SMEs, Malaysia | [81] |
| 3 | Do innovation and proactiveness impact the business performance of women-owned SMEs in Vietnam | 5 | 350 | SMEs, Vietnam | [82] |
| 4 | Using PLS-SEM technique to model construction organizations' willingness to participate in e-bidding | 6 | 64 | Clients and organizations | [76] |
| 5 | SEM-PLS analysis of inhibiting factors of cost performance for large construction projects in Malaysia: perspective of clients and consultants | 9 | 144 | Construction organizations, Malaysia | [83] |
| 6 | Blood production factors affecting transfusion sustainability: a study by using smart PLS-SEM approach | 3 | 241 | Blood bank, Uganda | [84] |
| 7 | A partial least squares structural equation modeling (PLS-SEM) of barriers to sustainable construction in Malaysia | 5 | 122 | construction stakeholders, Malaysia | [85] |
| 8 | How does social media use enhance employee's well-being and advocacy behaviour? Findings from PLS-SEM and fsQCA | 4 | 302 | Organizations, India | [79] |
| 9 | Investigating the moderating role of education on a structural model of restaurant performance using multi-group PLS-SEM analysis | 6 | 198 | Restaurants, Australia | [86] |
| 10 | Impact of capital structure and innovation on firm performance: Direct and indirect effects of capital structure | 3 | 220 | business owners, Mexico | [78] |
| 11 | Managers' attitudes and behavioral intentions towards using artificial intelligence for organizational decision-making: A study with Colombian SMEs | 8 | 83 | SMEs, Colombia | [87] |
| 12 | Data to model the prognosticators of luxury consumption: A partial least squares-structural equation modelling approach (PLS-SEM). | 4 | 157 | Consumers, South Africa | [77] |
| 13 | How can generative artificial intelligence improve digital supply chain performance in manufacturing firms? Analyzing the mediating role of innovation ambidexterity using hybrid analysis through CB-SEM and PLS-SEM | | 263 | manufacturing firms, Jordan | [88] |
| 14 | The effect of occupational moral injury on career abandonment intention among physicians in the context of the COVID-19 pandemic | | 201 | physicians | [89] |

Furthermore, diverse respondent types, ranging from SMEs and consumers to physicians and manufacturers, illustrate the flexibility and applicability of this method across research contexts.

Together with theoretical guidance from [70], this empirical benchmarking confirms that a sample of 354 is sufficient for generalization and structural modelling in PLS-SEM. Drawn from a population of 3,908 transportation SMEs using a stratified probability method, the 9% sample meets standards for external validity and aligns with practices in peer-reviewed organizational research, supporting the study's representativeness and generalizability.

### 3.4. Development of construct measures

This research examines two-way links between institutional factors and human capital in advancing sustainable transport, covering social, economic, and environmental aspects. Standard psychometric protocols were applied to secure measurement reliability and validity [90–91].

We first reviewed relevant literature to identify established measurement scales for each construct, then adapted them through two rounds of interviews with leaders of transportation enterprises. The initial phase included 60 interviews to explore themes surrounding institutions, human capital, and sustainable development. Insights from this stage informed a second round of 30 interviews aimed at clarifying unclear areas and ensuring the scales fit the HCMC context. Each 40–60 minute session was thematically analysed to confirm content validity and practical relevance. A Q-sort procedure [92–93] was then used, with participants ranking items for importance and clarity to refine and prioritise the list. Sustainable transportation development measures were drawn from [6], covering social, economic, and environmental aspects. Institutional factors were categorised as challenging (e.g., regulatory obstacles, corruption risks) or supporting (e.g., favorable regulatory framework). Human capital included leadership and motivation (see Table 2). The final questionnaire contained demographic items, 18 institutional factor questions, 12 human capital items, and 28 sustainable development items, all measured on 5-point Likert scales. Independent variables ranged from "Absolutely no association" to "Strong association," and dependent variables from "Absolutely disagree" to "Absolutely agree". A preliminary assessment with 50 managers from 18 districts confirmed scale reliability through Cronbach's alpha.

This systematic, context-specific approach ensures conceptual soundness while reflecting local realities, contributing to practical advances in sustainable transport development.

### 3.5. Evaluating the measurement structure

Reliability and validity checks were central to strengthening the measurement model. In line with PLS-SEM recommendations, internal consistency was examined using Cronbach's Alpha and Composite Reliability (CR), both surpassing the 0.70 benchmark [69–70]. Items with correlations to the total score under 0.30 were scrutinized and removed to enhance the scale's effectiveness [94]. Convergent validity was confirmed using standardized factor loadings (≥ 0.70) [69–70] and Average Variance Extracted values (≥ 0.50) [95]. Discriminant validity was assessed via the Fornell–Larcker criterion [95–96] and the Heterotrait – Monotrait (HTMT) ratio, with all HTMT values were below 0.85, confirming discriminant validity for distinct constructs [97]. Multicollinearity was excluded, as all VIF scores were under 5 [70]. Predictive capability in PLS-SEM was assessed via PLS-Predict with k-fold cross-validation, where $Q^2$\_predict values greater than zero indicated relevance [70].

Construct definitions and validity drew on prior studies and insights from two interview rounds with transportation enterprise executives (Table 2). This combination ensured the measurement tool was both statistically robust and contextually relevant to sustainable transport development.

### 3.6. Method Selection: Justification for PLS-SEM

**Theoretical and methodological alignment.** This study adopts a primarily exploratory orientation, aiming to build and expand theoretical insights into sustainable development within resource-constrained transportation enterprises. It

**Table 2. Overview of observed variables, their components, validation via literature and interviews, and corresponding sources.**

| Structure | Abbreviation (Observed Variable Component Name) | Factors from Literature and Managerial Input | % Agreement in Interviews | | Observed Variable Description | References |
|---|---|---|---|---|---|---|
| **Challenging Institutional Factor** (Comprising multiple subdimensions such as regulatory burdens, corruption, bureaucracy, lack of support) | **INN** | | Round 1 | Round 2 | | |
| Regulation-related constraints | INN1 (Component 1) | Regulations and laws are increasingly complex, with issues including unclear legal regulations and burdensome employment rules | 85 | 81 | Regulation complexity, unclear rules, burdensome employment. | [54,56,99] |
| Corruption-related challenges | INN2 (Component 2) | Unfair competition, corruption, and complicated licensing procedures | 78 | 75 | Unfair competition, corruption, licensing issues | [55,99,100] |
| Formal rules | INN3 (Component 3) | Uncertainty in business regulation laws. Heavy taxation | 72 | 68 | Regulatory uncertainty, high taxes. | [56,99,101] |
| Informal constraints | INN4 (Component 4) | Excessive administrative procedures, administrative charges, bureaucratic hurdles | 73 | 81 | Excessive procedures, charges, bureaucracy. | [55,100,101] |
| Support-related constraints | INN5 (Component 5) | Business support institutions do not understand firms concerns. Local lack of business support institutions. Business support institutions charge too high fees for their services. | 70 | 75 | Lack of support, high service fees. | [99] |
| Compliance-related limitations | INN6 (Component 6) | Challenges include inadequate infrastructure supply, environmental regulations, and foreign trade procedures. | 74 | 76 | Poor infrastructure, environmental, trade barriers. | [54,56,99] |
| **Supporting Institutional Factor** (Comprising multiple subdimensions such as formal institutions, cultural norms, informal trust systems) | **INP** | | | | | |
| Formal norms, political and judicial rules | INP1 (Component 1) | Property rights and contracts (among individuals, arising from the existence of scarce goods and pertaining to their use) reduce transaction costs, including those related to the creation, restructuring, and enforcement of institutions, as well as the costs incurred by market participants when using these institutions. | 92 | 94 | Property rights, transaction costs. | [46,102,103] |
| Cultural institutions | INP2 (Component 2) | Cultural institutions foster new businesses, enhance entrepreneurship, contribute to economic development, provide opportunities to make new contacts, and facilitate strong social connections. | 73 | 76 | Foster business, entrepreneurship, social connections. | [104] |
| Substitutive informal institutions | INP3 (Component 3) | Local micro-credit schemes and rotational cooperation in rural areas are compatible with formal institutions. | 69 | 71 | Micro-credit, rural cooperation. | [105] |

*(Continued)*

| Structure | Abbreviation (Observed Variable Component Name) | Factors from Literature and Managerial Input | % Agreement in Interviews | | Observed Variable Description | References |
|---|---|---|---|---|---|---|
| Informal institutions | INP4 (Component 4) | Trusted individuals and connections that engage in problem-solving, perform economic functions, and exert informal pressure on their members. | 73 | 75 | Trusted networks, economic functions. | [106,107] |
| Informal norms | INP5 (Component 5) | Informal norms emerged where individuals achieved goals through personal networks. | 95 | 94 | Personal network goals | [108] |
| Informal constraints (conventions) | INP6 (Component 6) | Trust-based networks can emerge in formal environments and can be generated either top-down or initiated bottom-up in organizations. | 64 | 61 | Trust networks, organizations. | [109,110] |
| **Human Capital** | **HU** | | | | | |
| Leadership | HU1 (Component 1) | Employees possess leadership skills and learn from each other. | 92 | 94 | Leadership skills, learning. | [10,111–113] |
| Motivation | HU2 (Component 2) | Employees consistently do their best, perform tasks with a lot of energy, and evaluate their actions | 83 | 87 | Effort, energy, self-evaluation. | |
| Qualifications | HU3 (Component 3) | Employees' competence is at a suitable level, and they are considered intelligent. | 70 | 73 | Competence, intelligence. | |
| Training | HU4 (Component 4) | The organization facilitates skill and qualification upgrades for employees as needed, and implements a training program for successors upon employee departure. | 79 | 81 | Skill support, successor training. | |
| Satisfaction | HU5 (Component 5) | Employees are satisfied with the organization, and the organization ensures it is getting the most from its employees. | 95 | 92 | Satisfaction, performance. | |
| Creativity | HU6 (Component 6) | Organization consistently generates new ideas | 93 | 97 | Idea generation | |
| **Social Development** | **SOD** | | | | | |
| Number of employees | SOD1 (Component 1) | Number of employees, improved employee, or community health and safety | 85 | 81 | Employees, health, safety | [6,7,53,114] |
| Social security costs | SOD2 (Component 2) | Social security costs, recognized and acted on the need to fund initiatives. | 91 | 93 | Social security, funding. | |
| Personnel and wages costs | SOD3 (Component 3) | Personnel expenses, wages, and benefits | 87 | 85 | Personnel costs detailed | [6,7,53] |
| Revenue per employee | SOD4 (Component 4) | Revenue generated per employee | 89 | 91 | Revenue per employee | |
| Labour productivity | SOD5 (Component 5) | Output per employee, productivity metrics | 75 | 70 | Labour productivity | |
| Gross value added per employee | SOD6 (Component 6) | Value added by each employee, economic contribution | 65 | 69 | Value added/ employee | |
| Average personnel costs | SOD7 (Component 7) | Average costs incurred per employee for salaries and benefits | 75 | 77 | Average personnel costs | |
| Employment growth rate | SOD8 (Component 8) | Rate of increase in employment levels | 81 | 79 | Excluded: Low standardized loadings | |
| Investment per employee | SOD9 (Component 9) | Investment allocated per employee | 60 | 58 | Investment/ employee | |
| **Economic Development** | **ECD** | | | | | |

*(Continued)*

**Table 2.** (Continued)

| Structure | Abbreviation (Observed Variable Component Name) | Factors from Literature and Managerial Input | % Agreement in Interviews | | Observed Variable Description | References |
|---|---|---|---|---|---|---|
| Number of enterprises | ECD1 (Component 1) | Total number of enterprises in the area | 89 | 87 | Number of enterprises. | [6,7,53] |
| Enterprise revenue | ECD2 (Component 2) | Total revenue generated by enterprises | 91 | 88 | Enterprise revenue. | |
| Value added by factor cost | ECD3 (Component 3) | Economic value added after factor costs | 93 | 92 | Value added. | |
| Total operating surplus | ECD4 (Component 4) | Excess of revenue over operating expenses | 75 | 74 | Operating surplus. | |
| Total expenditure on goods and services | ECD5 (Component 5) | Total spending on goods and services by enterprises | 82 | 78 | Expenditure | |
| Total investment in tangible assets | ECD6 (Component 6) | Total investment in physical assets | 70 | 68 | Tangible investment. | |
| Investment ratio | ECD7 (Component 7) | Investment ratio (investment/value added by factor cost) | 61 | 65 | Investment ratio | |
| Average personnel costs | ECD8 (Component 8) | Average costs incurred per employee for salaries and benefits | 65 | 62 | Excluded: VIF > 5 | |
| Share of personnel costs in operation | ECD9 (Component 9) | Proportion of total operational expenses attributed to personnel costs | 60 | 73 | Excluded: VIF > 5 | |
| **Environmental Development** | **END** | | | | | |
| Carbon dioxide (CO2) emissions | END1 (Component 1) | CO2 emissions levels | 80 | 73 | CO2 emissions | [6,7,53] |
| Methane emissions | END2 (Component 2) | Methane emissions levels | 92 | 89 | Methane emissions | |
| Nitrous oxide emissions | END3 (Component 3) | Nitrous oxide emissions (gas with a mild sweet smell) | 65 | 63 | Excluded: VIF > 5 | |
| Sulphur oxide emissions | END4 (Component 4) | Sulphur oxide emissions measurements | 89 | 85 | Sulphur oxide emissions. | |
| Carbon Monoxide Emissions | END5 (Component 5) | Carbon monoxide emissions (a toxic gas formed by burning carbon, especially in the form of automobile fuel) | 65 | 67 | Excluded: VIF > 5 | |
| Nitrogen dioxide emissions | END6 (Component 6) | Nitrogen dioxide emissions (a toxic brown gas) level | 73 | 70 | Nitrogen diox-ide emissions. | |
| Ammonia emissions | END7 (Component 7) | Ammonia emissions (strong-smelling gas) level | 78 | 81 | Ammonia emissions. | |
| Resource use optimization | END8 (Component 8) | Optimized resource utilization by reducing the use of traditional fuels and substituting them with less polluting energy sources, thereby reducing waste and emissions from operations. | 92 | 95 | Resource use, reduced emissions | [53,114,115] |
| Voluntary remediation actions | END9 (Component 9) | Remediation: Undertook voluntary actions (not required by regulation) to restore the environment. | 60 | 58 | Voluntary remediation | |
| Minimization of impacts on wildlife and habitats | END10 (Component 10) | Efforts to minimize harm to wildlife and natural habitats | 72 | 78 | Excluded: VIF > 5 | |

Note: The constructs and their observed variables components were developed through a rigorous literature review and further refined via expert interviews with transportation enterprise managers and academic scholars. Institutional factors are operationalized using multidimensional subscales grounded in institutional economics, capturing both formal and informal institutional dimensions. Each factor, challenging institutions (INN)/supporting institutions (INP), comprises several components and is not treated as a simple binary. Likewise, human capital (HU) is conceptualized as a multidimensional construct, including leadership, motivation, qualifications, training, satisfaction, and creativity. The creativity item reflects employees' capacity to generate ideas and contribute to firm-level innovation, in line with prior studies that recognize creativity as a precursor to innovation. The components of sustainable transportation development, comprising SOD, ECD, and END dimensions, are grounded in sustainable transport theory and contextualized through empirical studies within the urban transport sector.

is grounded in sustainable transportation theory [2,3,6,32], North's institutional theory [9,46], to account for both negative and positive institutional factors, and Becker's human capital theory [59] to construct the human capital dimension. While the study emphasizes exploration, it also incorporates confirmatory components through hypothesis testing and the validation of structural relationships. PLS-SEM is particularly appropriate for this exploratory-confirmatory hybrid approach [70]. In contrast to Covariance-Based SEM, PLS-SEM is well suited for smaller samples [70], accommodates both reflective and formative constructs [69–70], and estimates measurement and structural models concurrently. These advantages match the study's complexity, which involves six latent variables and more than 50 indicators.

**Relevance to transportation SMEs in challenging settings.** PLS-SEM is especially fitting for studies on SMEs in resource-limited, regulation-heavy environments. This fits the current research, which examines HCMC transportation SMEs facing significant institutional constraints and restricted human capital capacity. Recent applications in comparable settings include studies on artificial intelligence adoption in Colombian SMEs [87] and digital innovation in Malaysian logistics SMEs [81]. Additional research has used PLS-SEM to examine the determinants of enterprise risk management in Ghanaian SMEs [98] and the drivers of business performance in Vietnamese SMEs [82]. These cases highlight the method's relevance for detecting associated links in transportation research within emerging urban economies, the precise focus of this study.

**Why PLS-SEM over regression or CB-SEM.** OLS regression cannot handle latent constructs, assumes error-free indicators, and fails to capture interdependent relationships. CB-SEM, while statistically rigorous, demands large samples, multivariate normality, and rigid model specification, often unrealistic in applied contexts. PLS-SEM, by contrast, fits moderate sample sizes (target = 300), non-normal data, and exploratory aims [69–70]. In this study, constructs like institutional factors (supporting and challenging), human capital, and the three pillars of sustainable transport are abstract, measured via multiple indicators. PLS-SEM estimates measurement and structural models jointly, producing path coefficients, significance levels, and reliability metrics. However, as the study is based on cross-sectional survey data, the results reflect statistically significant associations rather than causal effects. It handles hierarchical or multidimensional constructs well and resists problems like non-normality and multicollinearity [69–70]. With six latent variables, each with at least six measures, PLS-SEM was not just suitable but necessary for methodological soundness. Tools like PLS-Predict further add applied relevance by confirming predictive strength.

### 3.7. Research ethics compliance

The study used non-invasive surveys and interviews with executives of transport firm in HCMC. The dataset contained no sensitive or personally identifiable information, and all adult participants provided voluntary consent after being informed of the research purpose. Verbal consent was obtained and confirmed through email or messaging. Under institutional and local guidelines, formal ethics approval was unnecessary for research limited to business-related interviews without gathering sensitive personal information.

## 4. Study results

### 4.1. Pilot test results

The pilot test, based on a sample of 50 respondents (Appendix S1), revealed that 70% were male and 76% held university degrees, with the largest age group being 36–45 years (56%) and 60% working as managers; 36% had 11–15 years of experience, and most respondents were employed by limited liability firms (50%) and had fewer than 11 employees (70%). Descriptive statistics showed mean values for observed variables ranging from 2.58 to 3.86, while standard deviations varied from 0.702 to 1.332, reflecting the central tendency and dispersion of the data [116]. Cronbach's alpha analysis (Appendix S2) initially excluded supporting institutions (component 6), social development (component 9), and environmental development (component 9) due to item-rest correlations under 0.3. Subsequent checks showed all correlations exceeded 0.3, with final reliability coefficients above 0.875 for all constructs [69–70].

## 4.2. Official survey data analysis and PLS-SEM results

**4.2.1. Participant profile and descriptive statistics.** Appendix S3 presents demographic information for the 354 valid respondents: 59% male and 41% female; 62% hold university degrees. The majority (42%) are aged 36–45. Managers account for 60% of participants, primarily from limited liability companies (42%) with smaller labour forces. Out of 620 distributed questionnaires, 372 were returned, with 354 considered valid, surpassing the target by 12%, the sample included 57.1% of profitable transportation firms. Mean scores for challenging institutions, supporting institutions, and human capital (independent variables), along with social, economic, and environmental development (dependent variables), ranged from 2.751 to 4.199, with standard deviations between 0.863 and 1.174. Summary data are shown in Table 3 and Fig 3; full statistics are in Table 4. These descriptive measures provide insights into central tendencies and variability [116].

**4.2.2. Measurement model and reliability assessment results.** To ensure construct robustness, multiple reliability and validity assessments were conducted. As shown in Table 4, internal consistency was confirmed with Cronbach's

**Table 3. Descriptive statistics analysis of the official survey.**

| Variable | N | Mean | Standard deviation | Min | Max |
|---|---|---|---|---|---|
| Challenging Institutions | 354 | 3.199 | 0.986 | 1 | 5 |
| Supporting Institutions | 354 | 4.199 | 1.100 | 1 | 5 |
| Human Capital | 354 | 3.729 | 0.863 | 1 | 5 |
| Social Development | 354 | 2.751 | 0.904 | 1 | 5 |
| Economic development | 354 | 3.452 | 1.174 | 1 | 5 |
| Environmental Development | 354 | 3.495 | 1.109 | 1 | 5 |

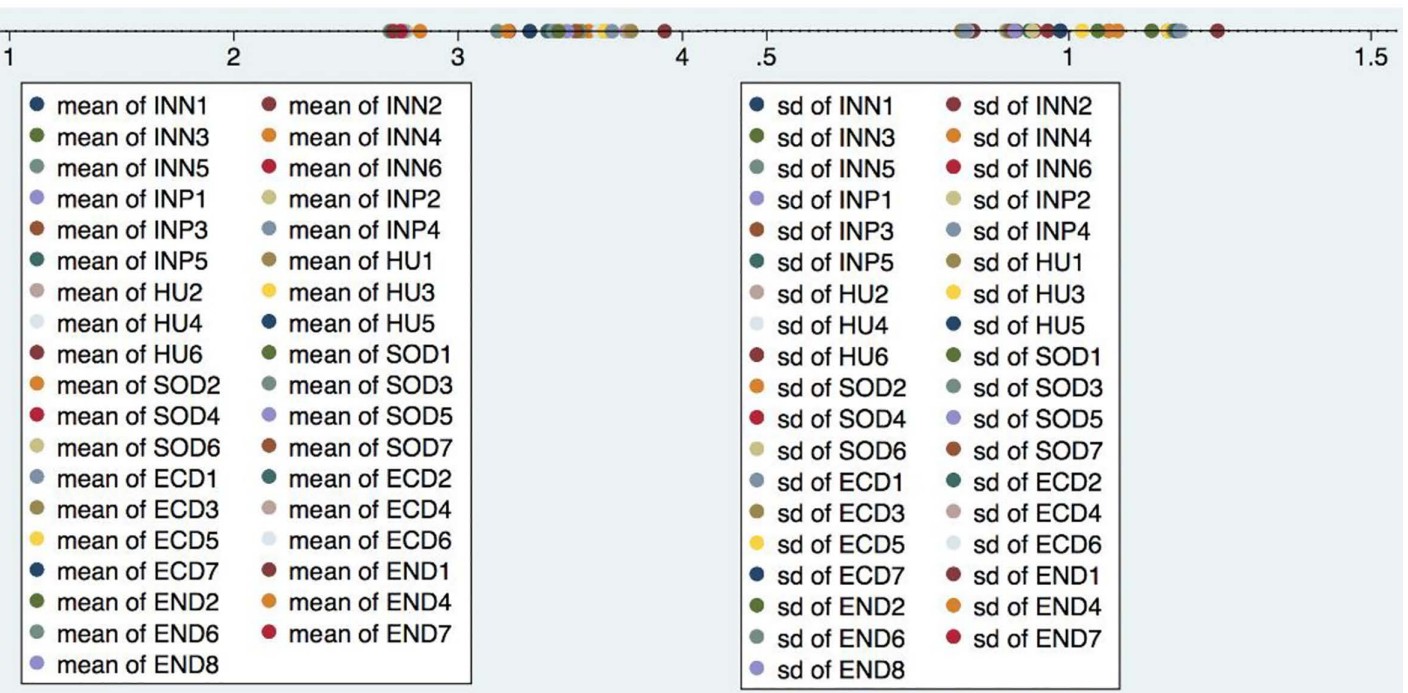

**Fig 3. Visual graph of descriptive statistics analysis of the official survey.**

**Table 4. Reliability and measurement model assessment.**

| Observed Variable Components | Abbre-viation | VIF | Standardized Loadings | AVE | Cronbach's Alpha | CR | (HTMT) |
|---|---|---|---|---|---|---|---|
| **Challenging Institutional factors are assumed to have a negative influence** | **INN** | | | 0.716 | 0.92 | 0.92 | 0.097 |
| Regulation complexity, unclear rules, burdensome employment | INN1 | | 0.874 | | | | |
| Unfair competition, corruption, licensing issues | INN2 | 1.94 | 0.798 | | | | |
| Regulatory uncertainty, high taxes | INN3 | 2.34 | 0.804 | | | | |
| Excessive procedures, charges, bureaucracy | INN4 | 2.65 | 0.847 | | | | |
| Lack of support, high service fees | INN5 | 2.96 | 0.878 | | | | |
| Poor infrastructure, environmental, trade barriers | INN6 | 2.04 | 0.874 | | | | |
| **Supporting Institutional factors are assumed to have a positive influence** | **INP** | | | 0.758 | 0.92 | 0.922 | 0.2 |
| Property rights, transaction costs | INP1 | | 0.819 | | | | |
| Foster business, entrepreneurship, social connections | INP2 | 2.67 | 0.868 | | | | |
| Micro-credit, rural cooperation | INP3 | 2.68 | 0.855 | | | | |
| Trusted networks, economic functions | INP4 | 3 | 0.882 | | | | |
| Personal network goals | INP5 | 3.73 | 0.925 | | | | |
| **Human Capital** | **HU** | | | 0.639 | 0.883 | 0.898 | 0.398 |
| Leadership skills, learning | HU1 | | 0.871 | | | | |
| Effort, energy, self-evaluation | HU2 | 2.99 | 0.879 | | | | |
| Competence, intelligence | HU3 | 1.48 | 0.75 | | | | |
| Skill support, successor training | HU4 | 2.13 | 0.799 | | | | |
| Satisfaction, performance | HU5 | 3.22 | 0.871 | | | | |
| Idea generation | HU6 | 1.28 | 0.587 | | | | |
| **Social development** | **SOD** | | | 0.726 | 0.936 | 0.938 | 0.183 |
| Employees, health, safety | SOD1 | | 0.764 | | | | |
| Social security, funding | SOD2 | 2.96 | 0.855 | | | | |
| Personnel costs | SOD3 | 2.21 | 0.793 | | | | |
| Revenue/employee | SOD4 | 3.34 | 0.879 | | | | |
| Productivity | SOD5 | 3.5 | 0.893 | | | | |
| Value added/employee | SOD6 | 2.92 | 0.876 | | | | |
| Personnel costs | SOD7 | 3.62 | 0.895 | | | | |
| Employment growth | SOD8 | 1.15 | | | | | |
| **Economic development** | **ECD** | | | 0.751 | 0.945 | 0.957 | 0.114 |
| Number of enterprises | ECD1 | | 0.85 | | | | |
| Enterprise revenue | ECD2 | 2.7 | 0.84 | | | | |
| Value added | ECD3 | 3.58 | 0.884 | | | | |
| Operating surplus | ECD4 | 2.68 | 0.832 | | | | |
| Expenditure | ECD5 | 3.35 | 0.886 | | | | |
| Tangible investment | ECD6 | 3.41 | 0.887 | | | | |
| Investment ratio | ECD7 | 3.07 | 0.883 | | | | |
| Personnel costs | ECD8 | 5.01 | | | | | |
| Personnel cost share | ECD9 | 5.2 | | | | | |
| **Environmental development** | **END** | | | 0.674 | 0.898 | 0.895 | 0.398 |
| CO2 emissions | END1 | | 0.845 | | | | |
| Methane emissions | END2 | 4 | 0.866 | | | | |
| Nitrous oxide emissions | END3 | 5.15 | | | | | |
| Sulphur oxide emissions | END4 | 3.98 | 0.853 | | | | |
| Carbon monoxide emissions | END5 | 5.34 | | | | | |

*(Continued)*

**Table 4.** (Continued)

| Observed Variable Components | Abbre-viation | VIF | Standardized Loadings | AVE | Cronbach's Alpha | CR | (HTMT) |
|---|---|---|---|---|---|---|---|
| Nitrogen dioxide emissions | END6 | 4 | 0.888 | | | | |
| Ammonia emissions | END7 | 3.27 | 0.884 | | | | |
| Resource use, reduced emissions | END8 | 1.09 | 0.533 | | | | |
| Minimized animal, habitat impact. | END10 | 5.27 | | | | | |

Note: VIF = Variance Inflation Factor; AVE = Average Variance Extracted; CR = Composite Reliability; HTMT = Heterotrait-Monotrait Ratio.

alpha values above 0.7 [69–70] for all constructs: challenging institutions (0.920), supporting institutions (0.922), human capital (0.883), and social (0.936), economic (0.945), and environmental development (0.898). Convergent validity was demonstrated as most standardized factor loadings exceeded 0.7, ranging from 0.750 to 0.925 for institutional and human capital indicators, and from 0.764 to 0.895 for social, economic, and environmental development constructs [69,70,95]. One item from human capital (component 6) and one from environmental development (component 8) fell slightly below 0.7 but were retained due to conceptual relevance and acceptable validity. CR scores further confirmed construct stability, exceeding 0.7 across all variables: 0.920 (challenging institutions), 0.922 (supporting institutions), 0.898 (human capital), and 0.895–0.957 (social, economic, and environmental development constructs) [69–70]. Average variance extracted values ranged from 0.639 to 0.758, surpassing the 0.50 threshold and confirming convergent validity [95–96]. Discriminant validity was confirmed via HTMT values, all well below the 0.85–0.90 cutoffs [97]. Multicollinearity was assessed using VIF; indicators with VIFs above 5 (ECD8, ECD9; END3, END5, END10) were removed, and the final model showed acceptable VIFs (1.093–1.501), indicating minimal multicollinearity [70]. Predictive validity was tested using PLS-Predict with 10-fold cross-validation. All key endogenous constructs had $Q^2$_predict values above zero, and most indicators showed lower RMSE than the linear benchmark, indicating medium to high predictive power [70]. These results confirm the model's reliability and validity in explaining relationships among institutions, human capital, and sustainable transportation development.

**4.2.3. Structural model analysis and hypotheses evaluation.** This section analyses the direct and reciprocal relationships between institutional factors and human capital with the social, economic, and environmental dimensions of sustainable transportation development. The proposed hypotheses were tested using PLS-SEM. Table 5 reports the R-squared ($R^2$) values, reflecting the model's explanatory capability. The average $R^2$ is 0.284, with moderate explanatory power for social dimension (0.323), environmental dimension (0.458), and human capital (0.459). Supportive institutions account for a smaller portion of variance (0.242), while challenging institutions (0.125) and economic dimension (0.061) show limited explanatory strength [70].

The results support H1(a1, a2, b1, b2, c1, c2), showing divergent institutional environment. Challenging institutions are negatively associated with social (β = − 0.195), economic (β = − 0.154), and environmental development (β = − 0.082). These negative associations are modest in size but statistically significant. According to [70], path coefficients closer to ±1 reflect stronger relationships; hence, − 0.082 is considered a weak association, while − 0.195 and − 0.154 represent small-to-moderate effects. In contrast, supporting institutions are positively associated with social (β = 0.328), economic (β = 0.120), and environmental development (β = 0.103). Among these, the positive associations with social development (0.328) approaches a moderate level, suggesting meaningful institutional practices. Although most of the negative associations are smaller in magnitude, they remain relevant and noteworthy.

Bidirectional relationships in H2(a1, a2, c1, c2) are supported. The social dimension is negatively associated with challenging institutions (β = − 0.250) and positively associated with supporting institutions (β = 0.388), both indicating moderate associations. Environmental development likewise exerts a negative association with challenging institutions (β = − 0.119)

**Table 5. Hypotheses Support and Structural Model Path Estimates (PLS-SEM).**

| Path | Direct Effect (β) | p-value | VIF | f² Value | Effect Size | Supported Hypotheses |
|---|---|---|---|---|---|---|
| INN→SOD | −0.195 | 0.000 | 1.093 | 0.0487 | Large | Yes – H1(a1) |
| INP→SOD | 0.328 | 0.000 | 1.203 | 0.1300 | Large | Yes – H1(a2) |
| INN→ECD | −0.154 | 0.005 | 1.093 | 0.0202 | Moderate to Large | Yes – H1(b1) |
| INP→ECD | 0.120 | 0.035 | 1.203 | 0.0106 | Medium | Yes – H1(b2) |
| INN→END | −0.082 | 0.047 | 1.093 | −0.0148 | (Negligible / not meaningful) | Yes – H1(c1) |
| INP→END | 0.103 | 0.017 | 1.203 | −0.0203 | (Negligible / not meaningful) | Yes – H1(c2) |
| SOD→INN | −0.250 | 0.000 | 1.501 | | | Yes – H2(a1) |
| SOD→INP | 0.388 | 0.000 | 1.501 | | | Yes – H2(a2) |
| ECD→INN | −0.093 | 0.083 | 1.151 | | | No – H2(b1) |
| ECD→INP | 0.020 | 0.691 | 1.151 | | | No – H2(b2) |
| END→INN | −0.119 | 0.034 | 1.349 | | | Yes – H2(c1) |
| END→INP | 0.173 | 0.001 | 1.349 | | | Yes – H2(c2) |
| HU→SOD | 0.255 | 0.000 | 1.192 | 0.0768 | Large | Yes – H3(a) |
| HU→ECD | 0.086 | 0.128 | 1.192 | | | No – H3(b) |
| HU→END | 0.609 | 0.000 | 1.192 | 0.5867 | Large | Yes – H3(c) |
| SOD→HU | 0.152 | 0.001 | 1.501 | | | Yes – H4(a) |
| ECD→HU | 0.001 | 0.982 | 1.151 | | | No – H4(b) |
| END→HU | 0.598 | 0.000 | 1.349 | | | Yes – H4(c) |

Note:

All coefficients are standardized path coefficients.

Significance threshold: $p < 0.05$.

R² values represent the proportion of variance explained in the endogenous constructs.

Average $R^2 = 0.284$ indicates a moderate explanatory power of the model overall [70].

R² (SOD) = 0.323, R² (ECD) = 0.061, R² (END) = 0.458, R² (INN) = 0.125, R² (INP) = 0.242, R² (HU) = 0.459

and a positive association with supporting institutions (β = 0.173). These coefficients, though smaller, remain statistically significant and suggest that transportation sustainability outcomes contribute to institutional dynamics, albeit with varying strengths.

Finally, H3(a, c) and H4(a, c) are confirmed. Human capital exhibits a strong positive association with environmental development (β = 0.609), indicating its substantial role in enhancing transportation sustainability outcomes. As explained by [70], a coefficient of 0.609 implies that a one standard deviation increase in human capital is associated with a 0.609 standard deviation increase in environmental development, controlling for other predictors. This represents a substantial relationship. The association with social development (β = 0.255) and reciprocal associations from social (β = 0.152) and environmental development (β = 0.598) back to human capital are also statistically significant and range from moderate to strong in relevance.

While these path coefficients support the hypothesized directions, they do not establish causality. Given the cross-sectional design, the results reflect significant associations rather than direct causal effects. Nonetheless, the magnitude of the standardized coefficients provides insights into the relative importance and practical implications of institutional factors and human capital in shaping sustainable transportation outcomes.

**Effect size (f²) evaluation.** In addition to reporting R² values and standardized path coefficients, effect sizes (f²) were calculated to evaluate the unique contribution of each exogenous construct to the endogenous latent variables, following the guidelines by [70]. The f² statistic is calculated as: $f^2 = (R^2_{included} - R^2_{excluded}) / (1 - R^2_{included})$, where $R^2_{included}$

is the R² value of the endogenous construct with the predictor included, and R²$_{excluded}$ is the R² value when the specific predictor is excluded from the model. Accordingly, three additional PLS-SEM models were estimated, each omitting one of the exogenous constructs, namely INN, INP, and HU, in order to isolate their effect on the endogenous variables, namely SOD, ECD, and END. Accordingly, three additional PLS-SEM models were estimated, each excluding one of the exogenous constructs, namely INN, INP, and HU, in order to isolate their individual effects on the endogenous variables SOD, ECD, and END. The R² values from these exclusion models are as follows:

When INN was excluded: Average R²=0.307; R² (SOD) = 0.290, R² (ECD) = 0.042, R² (END) = 0.466.

When INP was excluded: Average R²=0.276; R² (SOD) = 0.235, R² (ECD) = 0.051, R² (END) = 0.469.

When HU was excluded: Average R²=0.173; R² (SOD) = 0.271, R² (ECD) = 0.057, R² (END) = 0.140.

These results provide the necessary values for computing f² and allow a clearer interpretation of each construct's practical contribution to the explained variance in the model.

The resulting nine f² values are presented in Table 5. The interpretation follows [70], who report that Kenny (2018) suggests f² values of 0.005, 0.01, and 0.025 represent more realistic thresholds for small, medium, and large effect sizes in moderation analysis, while also cautioning that even these benchmarks may be optimistic. The strongest effect was observed for HU on END (f²=0.5867), indicating a large effect size. Notable effects were also found for INP on SOD (f²=0.1300), HU on SOD (f²=0.0768), and INN on SOD (f²=0.0487), all suggesting large effect sizes. A moderate effect was identified for INP on ECD (f²=0.0202). The effects of INN and INP on END (f²=− 0.0148 and − 0.0203, respectively) were negligible or not meaningful. These findings enhance the interpretation of the structural model by quantifying the practical significance of each predictor's unique contribution to the dimensions of sustainable transportation development.

## 5. Discussion

### 5.1. Key findings and scope of discussion

This study examined the statistical relationships among supportive and challenging institutional factors, human capital, and the social, economic, and environmental dimensions of sustainable transportation, using transportation enterprises in HCMC as a case. The findings substantiate several hypothesized relationships and offer clear empirical responses to the research questions.

Specifically, enabling institutional factors were positively associated with all three dimensions of sustainability, supporting hypotheses H1(a2), H1(b2), and H1(c2), whereas constraining institutional factors showed negative associations with these same dimensions, supporting H1(a1), H1(b1), and H1(c1). Notably, the results also reveal bidirectional associations. The social and environmental dimensions were positively associated with supportive institutions, supporting H2(a2) and H2(c2), and negatively associated with constraining institutions, supporting H2(a1) and H2(c1). These findings directly address the core research question and hypotheses regarding the dynamic interplay between institutional quality and sustainability outcomes in transportation enterprises. Human capital also emerged as a key factor in promoting sustainability. It was positively associated with both social and environmental sustainability, supporting H3(a) and H3(c), and in turn, both dimensions were positively associated with human capital, supporting H4(a) and H4(c). These dynamics reflect a mutually reinforcing system in which institutional conditions and human capital are interconnected through reciprocal associations, rather than functioning independently. While institutional factors may serve to enable or constrain sustainability, human capital acts as a key facilitator of sustainable outcomes within transportation enterprises.

It is essential to emphasize that these findings represent statistical associations rather than causal effects, given the study's cross-sectional design. Nevertheless, the effect size, ranging from small to large, highlight the practical significance of both institutional and human capital factors in shaping sustainable transportation outcomes.

In the specific context of HCMC, where institutional reform is evolving under a socialist-oriented market economy, the delayed development of formal institutions relative to economic growth reinforces the relevance of human capital and

informal institutional mechanisms. These dynamics offer theoretical and practical contributions to urban sustainability strategies in transitional economies.

### 5.2. Unexpected findings: Economic development's limited bidirectional association

One of the most significant and unexpected discoveries of this study concerns the limited role of economic development in the hypothesized bidirectional associations. Specifically, economic development was not significantly associated with either supporting or constraining institutional factors, thus not supporting H2(b1) and H2(b2), and did not exhibit a significant association with human capital, thereby not supporting H4(b). Furthermore, human capital was not found to be directly associated with economic development, thus not supporting H3(b). These findings differ from much of the existing literature, which often views economic growth as both a driver and outcome of institutional quality and human capital development.

Several contextual explanations may account for these null findings. As noted by [117], institutional quality is associated with economic performance indirectly, primarily through mediating factors such as infrastructure. This aligns with the possibility that institutional or human capital improvements alone may not yield measurable economic benefits unless accompanied by complementary developments in physical infrastructure or market access. In this case, the firm-level data from transportation enterprises may not fully capture such systemic interdependencies.

In the context of HCMC, rapid economic liberalization has often outpaced institutional reforms and the parallel development of human capital. This mismatch may produce conditions in which economic gains occur independently of institutional strength or workforce capability. Furthermore, Vietnam's increasing alignment with the United Nations' SDGs has shifted policy focus from pure economic expansion toward broader objectives including environmental sustainability and social equity. Similar trends are evident in other socialist-oriented economies, where accelerated market liberalization has outpaced the adaptive capacity of institutions [118–119].

Methodologically, while the sample size (n = 354) meets recommended standards for PLS-SEM [70], it is possible that more subtle or mediated relationships require either larger samples or longitudinal data to detect. In addition, self-reported measures, particularly on abstract constructs such as institutional trust, can be influenced by respondents' interpretation and reporting bias [120].

Overall, these findings invite further investigation into the complex and context-specific mechanisms that link economic development, institutional frameworks, and human capital, particularly in rapidly transforming urban economies. They also emphasize the importance of avoiding assumptions of universality in sustainability dynamics, and instead call for locally grounded, multidimensional analyses.

### 5.3. Model explanatory power and effect sizes

Model performance was assessed using $R^2$ values and effect sizes ($f^2$) based on PLS-SEM analysis [70]. The model exhibits moderate explanatory power for human capital ($R^2 = 0.459$), environmental development ($R^2 = 0.458$), and social development ($R^2 = 0.323$), highlighting institutional dynamics and workforce quality as key drivers of sustainable transportation outcomes. In contrast, the variance explained for supporting institutional factors ($R^2 = 0.242$) is lower, while economic development ($R^2 = 0.061$) and challenging institutional factors ($R^2 = 0.125$) demonstrate weak explanatory.

Effect sizes ($f^2$), computed through exclusion models, clarify each construct's unique contribution. Human capital demonstrated the strongest effect on environmental development ($f^2 = 0.5867$), while supporting institutions and human capital had notable effects on social development ($f^2 = 0.1300$ and $f^2 = 0.0768$, respectively), and a moderate effect was observed for supporting institutions on economic development ($f^2 = 0.0202$). In contrast, both supporting and challenging institutions exhibited negligible or non-meaningful effects on environmental development. These findings highlight the practical importance of human capital and institutional environments in advancing sustainability transitions within transportation enterprises.

## 5.4. Comparison with previous empirical studies

To contextualize the results, Table 6 contrasts this study's path coefficients with findings from prior international research. This comparison reinforces the validity of the results and highlights how institutional and human capital factors relate to sustainable transportation development, showing both common patterns and context-specific variations.

Table 6 reports the standardized coefficients (β) for direct relationships between institutional factors, human capital, and the social, economic, and environmental dimensions of sustainable transportation development. Following [70], coefficients closer to ±1 indicate stronger relationships, while those near zero suggest weaker effects. Thus, interpretation considers both statistical and practical significance.

This study finds that enabling institutions, coherent regulations, collaborative norms, and supportive informal structures, positively affect all sustainability dimensions: β = 0.328 (social), β = 0.120 (economic), and β = 0.103 (environmental). The social effect is moderately strong, while economic and environmental effects, though smaller, remain meaningful. These results align with findings from Australia [61], Central and Eastern Europe [6], and Western Europe [121], underscoring institutional support as a driver of sustainable transport. In contrast, constraining institutions, corruption, regulatory inefficiency, and misalignment, negatively affect sustainability: β = − 0.195 (social), β = − 0.154 (economic), and β = − 0.082 (environmental). Social and economic impacts are moderately negative; the environmental effect is weaker. Similar

**Table 6. Comparison of relationship coefficients with previous empirical studies.**

| Relationship | Social Dimension Coefficient | Economic Dimension Coefficient | Environmental Dimension Coefficient | Statistical Method | Study Location | Source |
|---|---|---|---|---|---|---|
| **Supporting Institutional Factor** (Effective formal rules, supportive norms, cooperative informal institutions) | 0.328*** | 0.12** | 0.103** | PLS-SEM | HCM City, Vietnam | This study |
| Legal compliance, eco-control, employee education, and environmental reporting | 0.049** | 0.049** | 0.491*** | Multiple Regression | Australia | [61] |
| Institutional policies for macroeconomic stability | 1.137–1.707*** | 1.137–1.707*** | 1.137–1.707*** | OLS Regressions | Bulgaria, Croatia, Estonia, Hungary | [6] |
| Economic growth and legal frameworks during COVID-19 | 0.438*** | 5.56e-07*** | 2.29e-06*** | OLS Regressions | France, Germany, Poland | [121] |
| **Challenging Institutional Factor** (Regulation burden, corruption, misaligned rules, high service fees) | −0.195*** | −0.154*** | −0.082** | PLS-SEM | HCM City, Vietnam | This study |
| Taxation and bribery effects | −1.249*** −0.285*** | | | Multiple Regression | Uganda | [122] |
| Bureaucracy, license issues, support service gaps | −0.394** to −0.044** | −0.394** to −0.044** | −0.394** to −0.044** | Regressions | Slovenia | [55] |
| **Human Capital** Leadership, learning, competence, employee satisfaction, *creativity* | 0.255*** | – | 0.609*** | PLS-SEM | HCM City, Vietnam | This study |
| Workplace improvements, education, and employee participation | 0.3*** | 0.3*** | 0.107** | Multiple Regression | Australia | [61] |
| Human capital in green logistics (safety, attitude, energy optimization) | 0.427*** | 0.427*** | 0.427*** | PLS-SEM | Bangladesh | [123] |
| Competence and advantage in transport services | 0.38*** | | | PLS-SEM | Indonesia | [124] |
| Broader human capital impact on transport enterprise performance | – | – | – | Quantitative & Economic Analysis | | [125] |

patterns are observed in Uganda [122] and Slovenia [55], where institutional weaknesses undermined transport sustainability efforts.

Human capital, leadership, creativity, competence, and employee satisfaction, shows a strong association: $\beta = 0.255$ (social) and $\beta = 0.609$ (environmental). The environmental effect is substantial per [70], highlighting human capital as pivotal in green transport and innovation. These results echo findings from Australia [61], Bangladesh [123], and Indonesia [124], as well as broader evidence [125] supporting human capital's transformative role.

While Table 6 confirms key direct relationships, it does not address reverse associations, how sustainability outcomes may affect institutional and human capital development, indicating a gap in empirical literature, particularly in emerging urban transport contexts.

## 5.5. Analysis of latent constructs via observed variables

This section explores the observed variable components underlying each latent construct, based on their standardized factor loadings (Tables 7–9). Following [70], loadings below 0.10 are very weak, 0.10–0.20 weak, 0.20–0.30 moderate, above 0.30 strong, and above 0.50 very strong. This assessment enhances understanding of how institutional and human capital factors relate to sustainable transportation development across social, economic, and environmental dimensions, highlighting key mechanisms that enable or constrain sustainability in urban transport enterprises within emerging economies.

Table 7 lists six primary components of challenging institutional factors. The greatest constraint is limited institutional support and high service costs (0.878), indicating weak government commitment and inefficiencies [99]. Complex regulations and unclear labour laws (0.874) create legal uncertainty [54–56]. Poor infrastructure and trade restrictions (0.874) reflect logistical and cross-border barriers [56,99]. Excessive bureaucracy (0.847) and regulatory unpredictability (0.804) illustrate systemic rigidity [54–55], while corruption and licensing problems (0.798) highlight informal institutional weaknesses [99–100]. These overlapping constraints raise compliance costs, reduce institutional trust, and hinder strategic innovation.

Table 8 outlines five enabling components of supporting institutions. The highest-loading factor is personal network goals (0.925), underscoring the role of relational trust [108]. Trusted networks (0.882) and a business-enabling environment (0.868) enhance enterprise adaptability [104,106,107]. Access to microcredit and rural cooperation (0.855) supports financial inclusion and resilience [105], while strong property rights and low transaction costs (0.819) indicate institutional efficiency [46,102,103]. These components show how institutional facilitation, both formal and informal, reduces uncertainty and promotes sustainable investment.

Table 9 presents six principal human capital components. Proactive personal energy and self-assessment (0.879) rank highest, emphasizing intrinsic motivation. Leadership and continuous learning (0.871) and job satisfaction (0.871) enhance resilience and performance. Mentorship and skill succession (0.799), competence (0.750), and creativity (0.587)

**Table 7. Challenging institutional factor components ordered by standardized loadings.**

| Challenging Institutional Factor | Description of Observed Variable Component | Standardized factor loadings |
|---|---|---|
| Component 5 | Lack of support, high service fees | 0.878 |
| Component 1 | Regulation complexity, unclear rules, burdensome employment | 0.874 |
| Component 6 | Poor infrastructure, environmental, trade barriers | 0.874 |
| Component 4 | Excessive procedures, charges, bureaucracy | 0.847 |
| Component 3 | Regulatory uncertainty, high taxes | 0.804 |
| Component 2 | Unfair competition, corruption, licensing issues | 0.798 |

**Table 8. Supporting institutional factor components ordered by standardized loadings.**

| Supporting Institutional Factor | Description of Observed Variable Component | Standardized factor loadings |
|---|---|---|
| Component 5 | Personal network goals | 0.925 |
| Component 4 | Trusted networks, economic functions | 0.882 |
| Component 2 | Foster business, entrepreneurship, social connections | 0.868 |
| Component 3 | Micro-credit, rural cooperation | 0.855 |
| Component 1 | Property rights, transaction costs | 0.819 |

**Table 9. Human capital components ordered by standardized loadings.**

| Human Capital | Description of Observed Variable Component | Standardized factor loadings |
|---|---|---|
| Component 2 | Effort, energy, self-evaluation | 0.879 |
| Component 1 | Leadership skills, learning | 0.871 |
| Component 5 | Satisfaction, performance | 0.871 |
| Component 4 | Skill support, successor training | 0.799 |
| Component 3 | Competence, intelligence | 0.750 |
| Component 6 | Idea generation | 0.587 |

reflect capacity development and innovation [10,111–113]. These results suggest that sustainable development relies not only on technical skills but also on embedded values and norms. In contexts where formal institutions are weak or fragmented, human capital plays a central role. Promoting self-directed learning, leadership, and creativity is crucial for supporting long-term sustainability in such environments.

## 5.6. Study contributions

This research advances theoretical insights from North's institutional theory by showing how both enabling and constraining institutional factors are associated with sustainable transportation development across social, economic, and environmental dimensions, and how these sustainability outcomes, in turn, are associated with institutional dynamics. Drawing on Becker's human capital theory, the study also examines how human capital is associated with, and reciprocally associated with, these same dimensions. Using HCMC as a case, an emerging urban transport hub with institutional complexity and workforce constraints, the findings highlight the role of both formal and informal institutions, whether supportive or obstructive, have direct impacts on sustainability outcomes. These institutional dynamics are reciprocally shaped, particularly by social and environmental progress.

Theoretically, this study extends North's view of institutions as evolving, co-adaptive systems shaped by societal and environmental transformations, and broadens Becker's framework by positioning human capital as central to social and environmental progress. A key contribution lies in the disaggregation of institutional and human capital constructs into observable components, offering more nuanced insights into their specific roles in sustainability transitions of transportation enterprises in constrained, transitional urban contexts.

Practically, the findings offer clear guidance for policymakers and transportation enterprise leaders in emerging economies. Supporting institutional elements, such as personal networks, relational trust, entrepreneurship support, and economic enabling functions, consistently promote sustainability across all three dimensions [46,102–108]. Conversely, challenging traits, such as weak institutional support, high service costs, legal ambiguity, poor infrastructure, and environmental barriers, impede sustainable progress [54–56,99,100]. Similarly, human capital attributes including creativity, self-assessment, leadership, continuous learning, performance, and succession planning strongly enhance social and

environmental outcomes [10,111–113]. These findings support integrated strategies combining institutional reform and human capital development, particularly in fragmented governance contexts.

The study's distinctive contribution lies in empirically demonstrating how institutional and human capital factors are not only associated with, but also reshaped by, sustainability outcomes in transportation enterprises. Unlike previous research that treats these relationships as linear or isolated, this study reveals their reciprocal, dynamic nature. By unpacking these constructs into actionable components, the research offers both theoretical refinement and a transferable framework applicable to other emerging urban settings navigating complex sustainability transitions.

## 6. Conclusion

This study examined how both enabling and constraining institutional factors, along with human capital, are associated with sustainable transportation development. Drawing on North's institutional theory and Becker's human capital theory, and using HCMC as a case, a major urban transport hub in an emerging economy marked by institutional complexity and resource constraints, the study reveals a dynamic system in which institutions and human capital are directly associated with sustainability and are, in turn, associated with it, particularly through social and environmental dimensions.

Supportive institutional factors are positively associated with sustainability across all dimensions, whereas constraining forces, especially limited policy support and high service costs, are negatively associated. Human capital shows a strong positive association with social and environmental development and is itself positively associated with improvements in these areas. In contrast, economic development does not show a significant association with institutional conditions or a direct connection with human capital, highlighting a potential gap between economic growth and broader sustainable development objectives in urban transport hubs in emerging economies.

### Practical implications

For transportation management teams: The results highlight the strategic value of investing in human capital. Enhancing workforce qualities such as effort, energy, self-assessment, leadership, continuous learning, and job satisfaction supports stronger environmental performance and social cohesion, both key aspects of sustainability. In complex regulatory contexts like HCMC, building internal capacity is vital for long-term resilience. Additionally, understanding and managing both enabling and constraining institutional factors, such as informal networks, administrative hurdles, infrastructure challenges, or regulatory uncertainty, helps transportation enterprises better align operational practices with sustainability goals. Integrated approaches that combine workforce development with institutional engagement and adaptive management are essential not only to drive progress but also to adapt to the evolving context, where sustainable development outcomes themselves are associated with both institutional conditions and human capital needs.

For investors and stakeholders: This study provides a data-driven framework for evaluating sustainability-oriented initiatives within transportation enterprises. By breaking down institutional and human capital factors, investors and stakeholders can more accurately evaluate enterprise-level capabilities associated with sustainability outcomes. In urban transport hubs in emerging economies, it is crucial to determine whether enterprises operate in supportive or fragmented institutional settings and whether they have the human capital to address environmental and social demands. Enterprises also need to adapt to evolving conditions in which sustainability outcomes are associated with both institutional arrangements and internal capacities. Targeted measures, such as workforce development initiatives or engagement with policy frameworks, can help overcome institutional barriers and create long-term value. Focusing investment decisions on environmental and social performance, rather than solely on economic growth, provides a clearer picture of sustainability potential in transitional urban contexts.

## 7. Study limitations and directions for future research

This study has several limitations. First, the initial phase included 60 interviews to identify broad patterns and themes from the survey data. A second round of 30 interviews was conducted to deepen qualitative insights; however, participation was limited by concerns over business confidentiality, reflecting institutional leadership's cautious approach to external research. Second, although the sample of 354 responses was sufficient for PLS-SEM analysis, the model showed that economic development is not significantly associated with institutions or human capital. This may reflect a context-specific gap between the economic dimension and both institutions and human capital, and also suggests that future research could benefit from using panel data. Third, due to software constraints, The PLS-SEM model was analysed in Stata 17, which does not support generating visualizations; therefore, Fig 4 was drawn manually and does not include path coefficients (β). Finally, this study employed a cross-sectional survey design, in which data were collected at a single point in time from each enterprise. While this design enables the identification of statistically significant associations between institutional factors, human capital, and transportation sustainability outcomes, it does not support causal inference. As such,

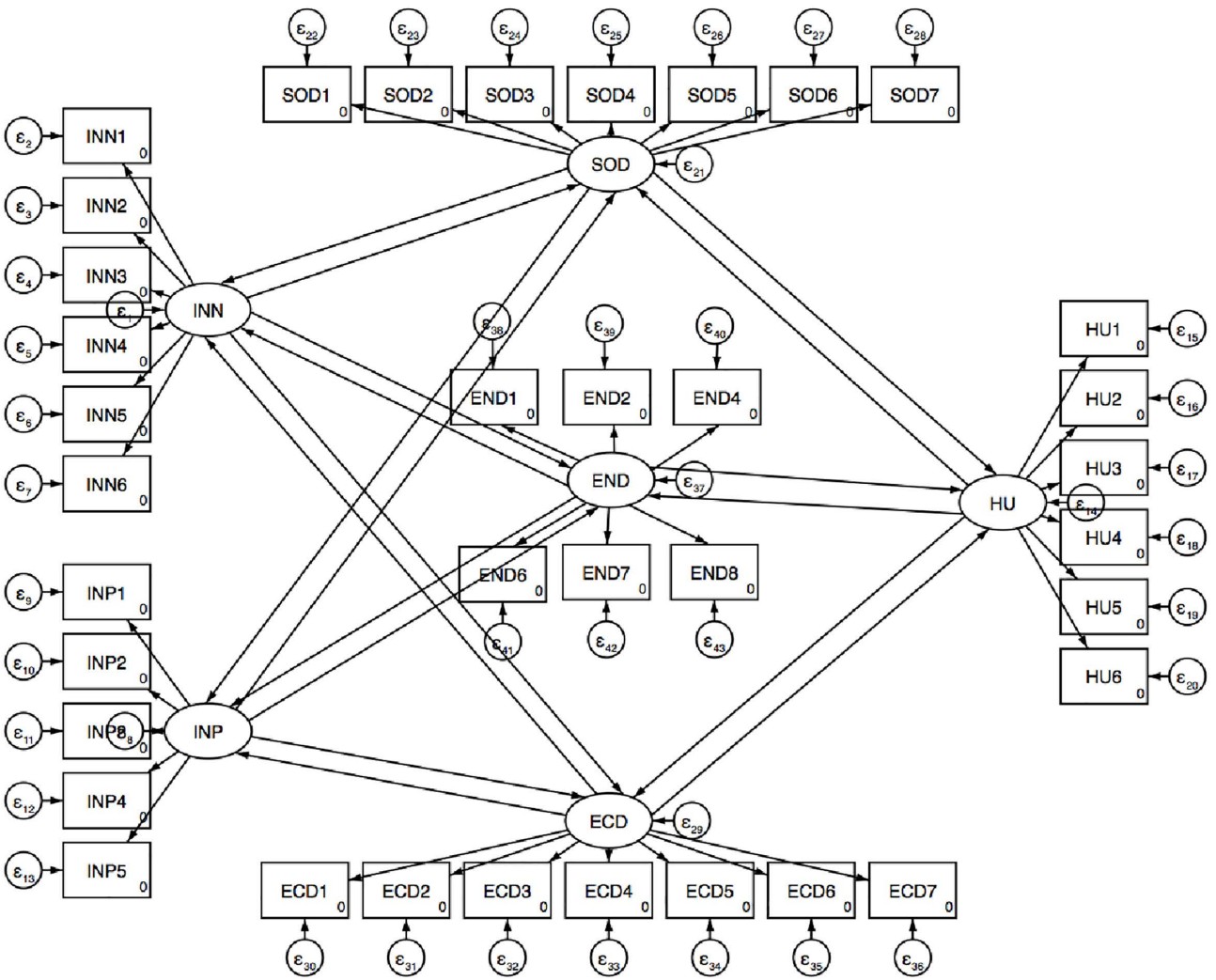

**Fig 4. PLS-SEM results visualization.**

the observed relationships are interpreted as associational rather than causal. Future research could enhance causal interpretation by adopting longitudinal or panel data to examine how these relationships evolve over time and respond to shifts in institutional and sustainable contexts.

## Supporting information

**S1 Appendix. Demographic and professional characteristics of pilot survey respondents.**
(DOCX)

**S2 Appendix. Cronbach's alpha test results of pilot test.**
(DOCX)

**S3 Appendix. Demographic and professional characteristics of official survey respondents.**
(DOCX)

**S1 File. Data file.**
(XLSX)

## Acknowledgments

We thank Dr. James Bleach & Dr. Susan Keron of the ōbex project for language editing support.

## Author contributions

**Conceptualization:** Vu Thi Kim Hanh, Nguyen Hong Nga.

**Data curation:** Vu Thi Kim Hanh.

**Formal analysis:** Vu Thi Kim Hanh.

**Funding acquisition:** Nguyen Hong Nga.

**Investigation:** Vu Thi Kim Hanh.

**Methodology:** Vu Thi Kim Hanh.

**Project administration:** Vu Thi Kim Hanh, Nguyen Hong Nga.

**Supervision:** Vu Thi Kim Hanh, Nguyen Hong Nga.

**Writing – original draft:** Vu Thi Kim Hanh.

**Writing – review & editing:** Vu Thi Kim Hanh.

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
