## [Decision Letter · Decision Letter 0]

2 Aug 2025

PONE-D-25-32321
Investigating How Institutions and Human Capital Influence Sustainable Transportation Development, and How It Responds in Return: Evidence from an Urban Transport Hub in an Emerging Economy
PLOS ONE

Dear Dr. Hanh,

Thank you for submitting your manuscript to PLOS ONE. After careful consideration, we feel that it has merit but does not fully meet PLOS ONE’s publication criteria as it currently stands. Therefore, we invite you to submit a revised version of the manuscript that addresses the points raised during the review process.

**ACADEMIC EDITOR:**

To enhance the quality of your manuscript, I encourage you to make revisions based on the feedback from our reviewers. Thank you.

We look forward to receiving your revised manuscript.

Kind regards,

Wong Ming Wong

Academic Editor

PLOS ONE

Journal Requirements:

2. In the online submission form, you indicated that the data that support the findings of this study are available from the corresponding author upon reasonable request.

"This research is funded by University of Economics and Law, Vietnam National University Ho Chi Minh City, Vietnam." We thank Dr. James Bleach & Dr. Susan Keron of the ōbex project for language editing support.

5. Please remove all personal information, ensure that the data shared are in accordance with participant consent, and re-upload a fully anonymized data set.

Additional Editor Comments 

To enhance the quality of your manuscript, I encourage you to make revisions based on the feedback from our reviewers. Thank you.

Reviewers' comments:

Reviewer's Responses to Questions

**Comments to the Author**

1. Is the manuscript technically sound, and do the data support the conclusions?

Reviewer #1: Yes

Reviewer #2: Partly

2. Has the statistical analysis been performed appropriately and rigorously? 

Reviewer #1: Yes

Reviewer #2: Yes

3. Have the authors made all data underlying the findings in their manuscript fully available?

Reviewer #1: Yes

Reviewer #2: No

4. Is the manuscript presented in an intelligible fashion and written in standard English?

Reviewer #1: Yes

Reviewer #2: No

5. Review Comments to the Author

Reviewer #1: Its well written but try to be short in writing

two shortcomings;

1. The author has divided the study according to the two theories. This creates an issue to naturally combine the theories in the end. Better if the author has discussed both of the theories in a debate form to draw a single list of hypotheses and research gaps,

2. The study is too long with a long list of references as well. It could have been squeezed.

Reviewer #2: Although the design and execution of the manuscript are technically sound, not all of the conclusions are entirely supported by the data as they stand at this time. Some theoretical concepts require more precise definitions, and statements regarding bidirectional influence need to be rephrased to account for cross-sectional SEM's limitations. The study might make a significant empirical contribution with the suggested changes in the attached file.

Please refer to the attached file for detailed comments on each of the Review Questions.

6. PLOS authors have the option to publish the peer review history of their article (what does this mean?). If published, this will include your full peer review and any attached files.

Reviewer #1: **Yes: **Dr. Khurshid Ahmad

Reviewer #2: **Yes: **Ajantha Kalyanaratne

---

## [Author Response · Author response to Decision Letter 1]

17 Aug 2025

Vu Thi Kim Hanh (Corresponding author)

Email: hanhvtk20702@sdh.uel.edu.vn

Nguyen Hong Nga (Co - corresponding author)

Email: nganh@uel.edu.vn

1. University of Economics and Law, Ho Chi Minh City, Vietnam.

2. Vietnam National University, Ho Chi Minh City, Vietnam.

17 August 2025

Editor-in-Chief

PLOS ONE

Dr Wong Ming Wong, Academic Editor

RESPONSE TO REVIEWER LETTER

Manuscript Number: PONE-D-25-32321

Previous title: Investigating How Institutions and Human Capital Influence Sustainable Transportation Development, and How It Responds in Return: Evidence from an Urban Transport Hub in an Emerging Economy

Revised title: Investigating Sustainable Development in Transportation Enterprises: Novel Insights from New Institutional Economics and Human Capital Theory. Evidence from HCM, Vietnam

Dear Dr Wong Ming Wong, Academic Editor, and Esteemed Reviewers,

We sincerely appreciate the opportunity to revise and resubmit our manuscript. We are grateful for the constructive feedback provided by you and the reviewers, which has guided us in strengthening the clarity, coherence, and overall quality of the manuscript. Your detailed insights have been invaluable in helping us refine both the conceptual framework and empirical presentation of our study.

As part of this resubmission, we would like to note a revision to the manuscript title. The title has been changed to “Investigating Sustainable Development in Transportation Enterprises: Novel Insights from New Institutional Economics and Human Capital Theory. Evidence from HCM, Vietnam” to ensure alignment with our officially registered research funding documentation. This adjustment does not affect the theoretical framework, methodology, or findings of the study, which remain fully consistent with the previously reviewed version. The revised title aims to maintain clarity while meeting institutional and funding requirements.

Below, we provide a detailed, point-by-point response to each reviewer comment. We address Reviewer 1's comments first, in the order they were presented, followed by responses to Reviewer 2’s comments. All revisions have been incorporated into the updated manuscript, with changes clearly marked using track changes for your review.

REVIEWER 1:

Reviewer 1’s Comment 1:

The author has divided the study according to the two theories. This creates an issue to naturally combine the theories in the end. Better if the author has discussed both of the theories in a debate form to draw a single list of hypotheses and research gaps,

Author’s Response to Reviewer 1 – Comment 1:

Thank you very much for this insightful and constructive suggestion. We fully agree that integrating the two theoretical perspectives more cohesively strengthens the conceptual coherence of the study. In response, we have revised the manuscript to present institutional theory and human capital theory in an integrated, debate-oriented form that highlights their interdependencies rather than treating them as separate analytical silos.

Specifically, we have a revised subsection titled “2.2. Integrating Institutional and Human Capital Theories: A Dual-Lens Approach to Sustainable Transportation Development” (see revised manuscript, pages 5 - 11; highlighted in blue and green). This section engages both theories in dialogue, emphasizing how institutional structures are associated with the sustainable transportation development, and also integrated with capital, and vice versa.

For instance, while North’s institutional theory addresses the formal and informal rules governing behaviour, Becker’s human capital theory explains how individuals interpret and respond to these institutional contexts. We elaborate on how enabling institutions (e.g., regulatory incentives, digital infrastructure) promote the development and retention of human capital, and how skilled personnel, in turn, reinforce or challenge institutional norms through innovation, leadership, and knowledge sharing. Conversely, constraining institutions (e.g., corruption, rigid bureaucracy) can limit the impact of even highly capable individuals. In return, empowered human capital can act as a catalyst for institutional improvement.

This conceptual integration directly informed our development of a unified research gap discussion, main contributions, and a consolidated set of hypotheses (see revised manuscript, pages 9 - 11). The hypotheses reflect the bidirectional associations between institutional factors, human capital, and the three dimensions of sustainable transportation development, economic, social, and environmental. We have structured them to capture both direct and reciprocal relationships, as recommended.

We sincerely hope this revised and integrated approach addresses your concern and enhances the theoretical coherence and contribution of the manuscript. Thank you again for your valuable guidance.

Reviewer 1’s Comment 2:

The study is too long with a long list of references as well. It could have been squeezed.

Author’s Response to Reviewer 1 – Comment 2:

We sincerely thank you for your valuable observation regarding the excessive length of the manuscript and the extensive reference list. In response, and in alignment with Reviewer 2’s related suggestions, we undertook a thorough and careful revision of the entire manuscript to improve conciseness and focus.

The revised manuscript now contains 13,208 words in total: 10,176 words in the main text (including author information) and 3,032 words in the references. For comparison, the previous version had 17,612 words in total, comprising 13,896 words (including Tables and author information) in the main text and 3,716 words in the references.

Reviewer 2 recommended that some of the tables, which were considered excessively detailed, be moved to supplemental material. After reviewing all nine tables, we decided to relocate all of them to the single Supporting Material file. This decision was made for the sake of consistency and to further enhance readability, ensuring that readers can focus more easily on the main findings and discussions.

Additionally, to maintain academic rigor and credibility, we carefully reviewed the reference list and removed 34 references that did not substantially contribute to the manuscript. This includes works from journals such as Sustainability, Atmosphere, and Energies (published by MDPI and listed on Beall’s List), as well as items not indexed in Scopus or Web of Science (e.g., J Trust and MIS Q, Lit Verlag, ITS & Transport Management Supplement). Sources of limited relevance were also excluded.

Instead, we supplemented the manuscript with several new, high-quality references to address specific points raised by Reviewer 2. These additional sources were carefully selected to strengthen the theoretical foundation, support the revised arguments, and ensure the scholarly robustness of the manuscript.

The list of removed references is provided below:

1. E. Macioszek, A. Granà, P. Fernandes, and M. C. Coelho, "New perspectives and challenges in traffic and transportation engineering supporting energy saving in smart cities—A multidisciplinary approach to a global problem," Energies, vol. 15, no. 12, Art. no. 4191, Jun. 2022. [Online]. Available: https://doi.org/10.3390/en15124191

2. Torosyan E, Amiryan S, Yesayan S. Development of human capital management system in the transportation industry. E3S Web Conf. 2020;164:10012. https://doi.org/10.1051/e3sconf/202016410012.

3. Ahuja S, Panigrahi BK, Dey N, Rajinikanth V, Gandhi TK. Deep transfer learning-based automated detection of COVID-19 from lung CT scan slices. Appl Intell 2021;51:571-85.

4. Franco S. The influence of the external and internal environments of multinational enterprises on the sustainability commitment of their subsidiaries: A cluster analysis. J Clean Prod 2021;297:126654. https://doi.org/10.1016/j.jclepro.2021.126654.

5. Schumacher AE, et al. Global age-sex-specific mortality, life expectancy, and population estimates in 204 countries and territories and 811 subnational locations, 1950–2021, and the impact of the COVID-19 pandemic: A comprehensive demographic analysis for the Global Burden of Disease Study 2021. The Lancet. 2024.

6. H. N. K. Vu et al., “Poor Air Quality and Its Association with Mortality in Ho Chi Minh City: Case Study,” Atmosphere, vol. 11, no. 7, p. 750, 2020, doi: https://doi.org/10.3390/atmos11070750.

7. H. T. Do, “Energy Demand Model Towards Sustainable On-Road Transportation in Ho Chi Minh City,” in Proceedings of the 8th International Conference on Sustainable Urban Development, V. T. Ha, H. N. Nguyen, and H.-J. Linke, Eds. Singapore: Springer Nature Singapore, 2024, pp. 303–320, https://doi.org/10.1007/978-981-99-8003-1_17.

8. Pojani D, Stead D. Sustainable urban transport in the developing world: Beyond megacities. Sustainability 2015;7(6):7784-805. https://doi.org/10.3390/su7067784.

9. Singh A, Gurtu A, Singh RK. Selection of sustainable transport system: a case study. Manag Environ Qual 2021;32(1):100-13.

10. Hietanen S. Mobility as a Service. The new transport model? ITS & Transport Management Supplement. 2014;2–4.

11. Kamargianni M, Matyas M, Li W, Schäfer A. A critical review of new mobility services for urban transport. Transp Res Procedia. 2016;14:3294–3303.

12. Shaheen SA, Chan ND, Gaynor T, & Camel T. Shared mobility: Definitions, industry developments, and early understanding. UC Berkeley: Transportation Sustainability Research Center; 2015.

13. Anderson JM, Nidhi K, Stanley KD, Sorensen P, Samaras C, Oluwatola OA. Autonomous Vehicle Technology: A Guide for Policymakers. Rand Corporation; 2016.

14. Goodall NJ. Machine ethics and automated vehicles. In: Meyer G, Beiker S, eds. Road Vehicle Automation. Springer International Publishing; 2014:93–102.

15. Bimbraw K. Autonomous cars: Past, present and future. Proc 12th Int Conf Inf Control Autom Robot (ICINCO). 2015;1(1):191–198.

16. Kucharcikova A, Miciak M. Human capital management in transport enterprises with the acceptance of sustainable development in the Slovak Republic. Sustainability 2018;10:2530. https://doi.org/10.3390/su10072530.

17. Kucharcikova A, Miciak M. Human capital management in transport enterprise. MATEC Web Conf 2017;134:1-6.

18. Willaby HW, Costa DSJ, Burns BD, MacCann C, Roberts RD. Testing complex models with small sample sizes: A historical overview and empirical demonstration of what partial least squares (PLS) can offer differential psychology. Pers Individ Differ. 2015;84:73–8.

19. Rindfleisch A, Malter AJ, Ganesan S, Moorman C. Cross-sectional versus longitudinal survey research: Concepts, findings, and guidelines. J Mark Res 2008;45(3):261–79.

20. Henseler J, Ringle CM, Sinkovics RR. The use of partial least squares path modeling in international marketing. In: Sinkovics RR, Ghauri PN, eds. New Challenges to International Marketing. Advances in International Marketing. Emerald Group Publishing Limited; 2009:277-19.

21. Hair JF, Ringle CM, Sarstedt M. PLS-SEM: Indeed a silver bullet. J Mark Theory Pract 2011;19:139-52.

22. G. Shmueli, S. Ray, J. M. Velasquez Estrada, and S. B. Chatla, “The elephant in the room: Predictive performance of PLS models,” Journal of Business Research, vol. 69, no. 10, pp. 4552–4564, Oct. 2016, doi: 10.1016/j.jbusres.2016.03.049.

23. J. F. Hair, G. T. M. Hult, C. M. Ringle, and M. Sarstedt, “The use of partial least squares structural equation modeling in strategic management research: A review of past practices and recommendations for future applications,” Long Range Planning, vol. 45, no. 5–6, pp. 320–340, 2012.

24. M. Sarstedt, C. M. Ringle, and J. F. Hair, “Beyond a tandem analysis of SEM and PROCESS: Use PLS-SEM for mediation analyses!,” *Int. J. Market Res.*, vol. 62, no. 3, pp. 288–299, 2020.

25. G. Cho and J. Y. Choi, “An empirical comparison of generalized structured component analysis and partial least squares path modeling under variance-based structural equation models,” *Behaviormetrika*, vol. 47, pp. 243–272, 2020.

26. J. F. Hair and M. Sarstedt, “Explanation plus prediction—The logical focus of project management research,” *Project Manag. J.*, vol. 52, no. 4, pp. 321–329, 2021.

27. J. F. Hair, J. J. Risher, M. Sarstedt, and C. M. Ringle, “When to use and how to report the results of PLS-SEM,” European Business Review, vol. 31, no. 1, pp. 2–24, 2019. doi: 10.1108/EBR-11-2018-0203.

28. Lang A, Murphy H. Business and sustainability. Cham: Springer; 2014. https://doi.org/10.1007/978-3-319-07239-5.

29. Orzeszyna K, Tabaszewski R. The legal aspects of activities taken by local authorities to promote sustainable development goals: Between global and regional regulations in Poland. Lex Localis 2021;19(1):1–20.

30. Ledeneva A. Informal networks in post-communist economies: A “topographic map.” In: Lahusen T, Solomon PH Jr., editors. What is Soviet Now? Identities, legacies, memories. Berlin: Lit Verlag; 2008. p. 101-34.

31. Möllering G. Trust, calculativeness, and relationships: A special issue 20 years after Williamson's warning. J Trust 2014;4(1):1–21.

32. G. Shmueli and O. R. Koppius, “Predictive analytics in information systems research,” *MIS Q.*, vol. 35, no. 3, pp. 553–572, 2011.

33. E. E. Rigdon, “Rethinking partial least squares path modeling: In praise of simple methods,” Long Range Plann., vol. 45, no. 5–6, pp. 341–358, 2012.

34. J. F. Hair Jr., G. T. M. Hult, C. M. Ringle, M. Sarstedt, N. P. Danks, and S. Ray, "An Introduction to Structural Equation Modeling," in Partial Least Squares Structural Equation Modeling (PLS-SEM) Using R, Cham, Switzerland: Springer, 2021, pp. 1–29, doi: 10.1007/978-3-030-80519-7_1

REVIEWER 2: 3) Major concerns

Reviewer 2: 1. Using PLS-SEM excessively without adequate model justification. The manuscript occasionally suggests confirmatory analysis, even though PLS-SEM is suitable for exploratory theory-building. Whether this study is exploratory or confirmatory needs to be made clear by the authors.

Authors response:

Thank you for this important observation. We have clarified the methodology section in 3.6. Method Selection: Justification for PLS-SEM (Theoretical and methodological alignment), on page 17 (highlighted in yellow). Specifically, we now explain that the study adopts a primarily exploratory approach grounded in theory-building, while also incorporating confirmatory elements through hypothesis testing. This justifies the use of PLS-SEM as an exploratory-confirmatory hybrid approach, supported by Hair et al. (2022)/[70].

Reviewer 2: For complete transparency, include measurement model tables (loadings, AVE, CR, and HTMT) in the main body of the text or as supplementary materials.

Authors' Response:

Thank you for your valuable comment. We appreciate your emphasis on ensuring transparency in reporting the measurement model.

In the revised manuscript, we confirm that the measurement model statistics, including standardized loadings, Average Variance Extracted (AVE), Cronbach's Alpha, Composite Reliability (CR), and Heterotrait-Monotrait Ratio (HTMT) values, are already presented in Table 4. The table is titled " Table 4. Reliability and Measurement Model Assessment" and is located in the single Supporting Material file.

The related explanatory text is provided in the revised manuscript under Section 4.2.2. Measurement Model and Reliability Assessment Results, on pages 20, highlighted in green for ease of review. The revised text in Section 4.2.2 clearly details the reliability and validity results, confirming that all constructs meet or exceed established statistical thresholds

To improve clarity and consistency, we have adjusted the abbreviations used in Table 4. Specifically, we now use EVA, CR, and HTMT in the column headings as follows:

Observed Variable Components | Abbreviation | VIF | Standardized Loadings | EVA | Cronbach's Alpha | CR | HTMT

Additionally, we have included a note at the end of Table 4 to define all abbreviations for ease of interpretation (see, page 21, highlighted in yellow): Note: VIF = Variance Inflation Factor; AVE = Average Variance Extracted; CR = Composite Reliability; HTMT = Heterotrait-Monotrait Ratio.

Once again, thank you for your constructive input and for helping us enhance the quality an

---

## [Decision Letter · Decision Letter 1]

15 Sep 2025

Investigating Sustainable Development in Transportation Enterprises: Novel Insights from New Institutional Economics and Human Capital Theory. Evidence from HCM, Vietnam

PONE-D-25-32321R1

Dear Dr. Hanh,

We’re pleased to inform you that your manuscript has been judged scientifically suitable for publication and will be formally accepted for publication once it meets all outstanding technical requirements.

Kind regards,

Wong Ming Wong

Academic Editor

PLOS ONE

Additional Editor Comments (optional):

Reviewer #2:

Reviewers' comments:

Reviewer's Responses to Questions

**Comments to the Author**

1. If the authors have adequately addressed your comments raised in a previous round of review and you feel that this manuscript is now acceptable for publication, you may indicate that here to bypass the “Comments to the Author” section, enter your conflict of interest statement in the “Confidential to Editor” section, and submit your "Accept" recommendation.

Reviewer #2: All comments have been addressed

2. Is the manuscript technically sound, and do the data support the conclusions?

Reviewer #2: Yes

3. Has the statistical analysis been performed appropriately and rigorously? 

Reviewer #2: Yes

4. Have the authors made all data underlying the findings in their manuscript fully available?

Reviewer #2: Yes

5. Is the manuscript presented in an intelligible fashion and written in standard English?

Reviewer #2: Yes

6. Review Comments to the Author

Reviewer #2: Thank you, authors, for addressing my review comments. I am satisfied with your responses. Please see the attached document for a detailed observations.

7. PLOS authors have the option to publish the peer review history of their article (what does this mean?). If published, this will include your full peer review and any attached files.

Reviewer #2: **Yes: **Ajantha Kalyanaratne

---

## [Editor Report · Acceptance letter]

PONE-D-25-32321R1

PLOS ONE

Dear Dr. Hanh,

I'm pleased to inform you that your manuscript has been deemed suitable for publication in PLOS ONE. Congratulations! Your manuscript is now being handed over to our production team.

Kind regards,

on behalf of

Dr. Wong Ming Wong

Academic Editor

PLOS ONE